# Synthetic vs Real: Deep Learning on Controlled Noise

## Abstract

Performing controlled experiments on noisy data is essential in thoroughly understanding deep learning across a spectrum of noise levels. Due to the lack of suitable datasets, previous research have only examined deep learning on controlled synthetic noise, and real-world noise has never been systematically studied in a controlled setting. To this end, this paper establishes a benchmark of real-world noisy labels at 10 controlled noise levels. As real-world noise possesses unique properties, to understand the difference, we conduct a large-scale study across a variety of noise levels and types, architectures, methods, and training settings. Our study shows that: (1) Deep Neural Networks (DNNs) generalize much better on real-world noise. (2) DNNs may not learn patterns first on real-world noisy data. (3) When networks are fine-tuned, ImageNet architectures generalize well on noisy data. (4) Real-world noise appears to be less harmful, yet it is more difficult for robust DNN methods to improve. (5) Robust learning methods that work well on synthetic noise may not work as well on real-world noise, and vice versa. We hope our benchmark, as well as our findings, will facilitate deep learning research on noisy data.

## 1 Introduction

Deep Neural Networks (DNNs) trained on noisy data demonstrate intriguing properties. For example, DNNs are capable of memorizing completely random training labels but generalize poorly on clean test data (Zhang et al., 2017). When trained with stochastic gradient descent, DNNs learn patterns first before memorizing the label noise (Arpit et al., 2017). These findings inspired recent research on noisy data. As training data are usually noisy, the fact that DNNs are able to memorize the noisy labels highlights the importance of deep learning research on noisy data.

To study DNNs on noisy data, previous work often performs controlled experiments by injecting a series of synthetic noises into a well-annotated dataset. The noise level $p$ may vary in the range of 0%-100%, where $p = 0\%$ is the clean dataset whereas $p = 100\%$ represents the dataset of zero correct labels. The most commonly used noise in the literature is uniform (or symmetric) label-flipping noise, in which the label of each example is independently and uniformly changed to a random (incorrect) class with probability $p$. Controlled experiments on noise levels are essential in thoroughly understanding a DNN's properties across a spectrum of noise levels and faithfully comparing the strengths and weaknesses of different methods. The synthetic noise enables researchers to experiment on controlled noise levels, and drives the development of theory and methodology in this field. On the other hand, some studies were also verified on real-world noisy datasets, *e.g.* on WebVision (Li et al., 2017a), Clothing-1M (Xiao et al., 2015), Fine-grained Images (Krause et al., 2016), and Instagram hashtags (Mahajan et al., 2018), where the images are automatically tagged with noisy labels according to their surrounding texts. However, these datasets do not provide true labels for the training images. Their underlying noise levels are not only fixed but also unknown, rendering them infeasible for controlled studies on noise levels. In this paper, we refer image-search noise in these datasets as "real-world noise" to distinguish it from synthetic label-flipping noise.

To study real-world noise in a controlled setting, we establish a benchmark of controlled real-world noisy labels, building on two existing datasets for coarse and fine-grained image classification: Mini-ImageNet (Vinyals et al., 2016) and Stanford Cars (Krause et al., 2013). We collect noisy labels using text-to-image and image-to-image search via Google Image Search. Every training image is

independently annotated by 3-5 workers, resulting in a total of 527,489 annotations over 147,108 images. We create ten different noise levels from $0\%$ to $80\%$ by gradually replacing the original images with our annotated noisy images. Our new benchmark will enable future research on the real-world noisy data with a controllable noise level.

We find that real-world noise possesses unique properties in its visual/semantic relevance and underlying class distribution. To understand the differences, we conduct a large-scale study comparing synthetic noise, namely blue-pilled noise (or *Blue noise*), and real-world noise (or *Red noise*[1]). Specifically, we train DNNs across 10 noise levels, 7 network architectures, 6 existing robust learning methods, and 2 training settings (fine-tuning and training from random initialization).

Our study reveals several interesting findings. First, we find that DNNs generalize much better on real-world noise than synthetic noise. Our results verify Zhang et al. (2017)'s finding of deep learning generalization on synthetic noise. However, we observe a considerably smaller generalization gap on real-world noise. This does not mean that real-world noise is easier to tackle. On the contrary, we find that real-world noise is more difficult for robust DNNs to improve. Second, our results substantiate Arpit et al. (2017)'s finding that DNNs learn patterns first on noisy data. But we find this behavior becomes insignificant on real-world noise and completely disappears on the fine-grained classification dataset. This finding lets us rethink the role of "early stopping" (Yao et al., 2007; Arpit et al., 2017) on real-world noisy data. Third, we find that when networks are fine-tuned, ImageNet architectures generalize well on noisy data, with a correlation of $r = 0.87$ and $0.89$ for synthetic and real-world noise, respectively. This finding generalizes Kornblith et al. (2019)'s finding, *i.e.* ImageNet architectures generalize well across clean datasets, to the noisy data.

Our contribution is twofold. First, we establish a large benchmark of controlled real image search noise. Second, we conduct perhaps the largest study in the literature to understand DNN training across a wide variety of noise levels and types, architectures, methods, and training settings. We hope our benchmark along with our findings, resulted from a considerable amount of manual labeling effort ($\sim$520K annotations) and computing resources ($\sim$3K experiments), will facilitate future deep learning research on real-world noisy data. Our main findings are summarized as follows:

1. DNNs generalize much better on real-world noise than synthetic noise. Real-world noise appears to be less harmful, yet it is more difficult for robust DNN methods to improve.

2. DNNs may not learn patterns first on the real-world noisy data.

3. When networks are fine-tuned, ImageNet architectures generalize well on noisy data.

4. Adding noisy examples to a clean dataset may improve performance as long as the noise level is below a certain threshold (30% in our experiments).

## 2 RELATED WORK

**Noisy Datasets**: to understand deep learning's properties on noisy training data, research often conducted experiments across a series of levels of synthetic noises. The most common one is uniform label-flipping noise (*aka.* symmetric noise), in which the label of each example is independently and uniformly changed to a random (incorrect) class with a probability (Zhang et al., 2017; Arpit et al., 2017; Vahdat, 2017; Shu et al., 2019; Jiang et al., 2018; Ma et al., 2018; Han et al., 2018; Li et al., 2019; Arazo et al., 2019). The synthetic noisy dataset enables us to experiment on controlled noise levels, and drive the development of theory and methodology in this field. Research have also examined other types of noise to better approximate the real-world noise distribution, including class-conditional noises (Patrini et al., 2017; Rolnick et al., 2017), noises from other datasets (Wang et al., 2018), *etc*. However, these noises are still synthetic, generated from artificial distributions. Furthermore, different types of synthetic noises may lead to inconsistent or even contradicting observations. For example, Rolnick et al. (2017) experimented on a slightly different type of uniform noise and surprisingly found that DNNs are robust to massive label noise.

On the other hand, studies have also verified DNNs on real-world noisy datasets. While other noise types exist, *e.g.* image omission and registration noise (Mnih & Hinton, 2012) or image corruption (Hendrycks & Dietterich, 2019), the most common type consists of images that are automatically tagged according to their surrounding texts either by directly crawling the web pages,

---

[1]Quoted from the Red and Blue pill in the movie "The Matrix (1999)".

*e.g.* Clothing-1M (Xiao et al., 2015), Instagram (Mahajan et al., 2018), or by querying an image search engine, *e.g.* WebVision (Li et al., 2017a). Several studies have used these datasets. For example, Guo et al. (2018), Jiang et al. (2018) and Song et al. (2018) verified their model on WebVision. Mahajan et al. (2018) trained large DNNs on noisy Instagram hashtags. As these datasets did not provide true labels for the training examples, methods could only be tested on a fixed and, moreover, unknown noise level. To the best of our knowledge, there have been no studies focused on investigating real noisy labels in a controlled setting. The closest work to ours is annotating a small set for evaluation (Veit et al., 2017) or estimating the noise level of the training data (Krause et al., 2016).

**Robust Deep Learning Methods**: robust learning is experiencing a renaissance in the deep learning era. Since training data usually contain noisy examples, the ability of DNNs to memorize all noisy training labels often leads to poor generalization on the clean test data. Recent contributions based on deep learning handled noisy data in multiple directions including, *e.g.*, dropout (Arpit et al., 2017) and other regularization techniques (Azadi et al., 2016; Noh et al., 2017), label cleaning/correction (Reed et al., 2014; Goldberger & Ben-Reuven, 2017; Li et al., 2017b; Veit et al., 2017), example weighting (Jiang et al., 2018; Ren et al., 2018; Shu et al., 2019; Jiang et al., 2015; Liang et al., 2016), semi-supervised learning (Hendrycks et al., 2018; Vahdat, 2017), data augmentation (Zhang et al., 2018; Cheng et al., 2019), *etc*. Few studies have systematically compared these methods across different noise types and training conditions. In our study, we select and compare six methods from four directions: regularization, label learning, example weighting, and mixup augmentation. See Section 4 for details. These examined methods are selected because they (i) represent a reasonable coverage of different ways of handling noisy data; (ii) are comparable to the state-of-the-art on the commonly used CIFAR-100 with synthetic noise.

## 3 DATASETS

Our benchmark is built on two existing datasets: Mini-ImageNet (Vinyals et al., 2016) for coarse image classification and Stanford Cars (Krause et al., 2013) for fine-grained classification. Mini-ImageNet provides images of size 84x84 with 100 classes from the ImageNet dataset (Deng et al., 2009)[2]. We select 50,000 images for training and 5,000 from ImageNet for testing. Note that unlike few-shot learning, we train and test on the same 100 classes. The Stanford Cars contain 16,185 high-resolution images of 196 classes of cars (Make, Model, Year) splitting in a 50-50 into training and test set. The standard test split is used.

To recap, let us revisit the construction of existing noisy datasets in the literature. For the real-world noisy datasets (Xiao et al., 2015; Li et al., 2017a), one automatically collects images for a class by matching the class name to the images' surrounding text (by web crawling or equivalently querying the crawled index). The retrieved images include false positive (or noisy) examples, *i.e.* text match/search thinks an image is a positive when it is not. As their training images are not manually labeled, the data noise level is fixed and unknown. As a result, these datasets are unsuitable for controlled studies.

On the other hand, the synthetic noisy dataset is built on the well-labeled dataset. The label of each training example is independently changed to a random incorrect class[3] with a probability $p$, called noise level, which indicates the percentage of training examples with false labels. Since the ground-truth labels for every image are known, previous studies enumerate $p$ to obtain datasets of different noise levels and use them in controlled experiments. On balanced datasets, such as Mini-ImageNet and Stanford Cars used in our study, the above process is equivalent to first sampling $p\%$ training images from a class and then replacing them with the images uniformly drawn from other classes. The drawback is that their noisy labels are artificial and do not follow the distribution of the real-world noise (Xiao et al., 2015; Li et al., 2017a; Krause et al., 2016).

For our datasets, we follow the construction of synthetic datasets with only one difference, *i.e.* we draw false positive (noisy) examples from similar noise distributions as in existing real-world noisy datasets (Li et al., 2017a; Xiao et al., 2015). To be specific, we draw images using Google

---

[2]We choose Mini-ImageNet over CIFAR for its larger image size and ability to fit in the memory of GPUs.

[3]This is slightly different from (Zhang et al., 2017) and is same as (Jiang et al., 2018). We do not allow examples to be label flipped to their true labels. It makes $p$ denote the exact noise level and also independent of the total number of classes.

text-to-image search, which is commonly used to get noisy labels in prior works (Bootkrajang & Kabán, 2012; Li et al., 2017a; Krause et al., 2016; Chen & Gupta, 2015; Wang et al., 2014). In addition, we also include noisy examples using image-to-image search to enrich the type of label noises in our dataset. We manually annotate every retrieved image to identify the ones with false labels. For each class, we replace $p\%$ training images in the Mini-ImageNet and Stanford Cars datasets with these false-positive images. We enumerate $p$ in 10 different levels: $\{0\%, 5\%, 10\%, 15\%, 20\%, 30\%, 40\%, 50\%, 60\%, 80\%\}$ to study noisy labels in the controlled setting. Since Mini-ImageNet and Stanford Cars are all collected from the web images, their true positive examples should follow a similar distribution as the added false positive images. The constructed datasets hence contain label noise similar to existing real-world datasets and are suitable to be used in controlled experiments.

We sample noisy images using Google Image Search[4] in three steps: images collection, deduplication, and annotation. In the first step, we combine images independently retrieved from two sources (1) text-to-image and (2) image-to-image search. For text-to-image (or text-search in short), we formulate a text query for each class using its class name and broader category (*e.g.* cars) to retrieve the top 5,000 images. For image-to-image search (or image-search), we query the search engine using every training image in Mini-ImageNet (50,000) and Stanford Cars (8,144). This collects a large pool of similar images. As different query images may retrieve the same image, we rank the retrieved images by their number of occurrences in the pool and remove images that occur less than 3 times. Finally, we union text-search and image-search images, where the text-search images accounts for 72% of our final dataset. For deduplication, following (Kornblith et al., 2019), we run a CNN-based duplicate detector over all images to remove near-duplicates to any of the images in the test set. All images are under the usage rights "free to use or share" [5].

These images are then annotated on a cloud labeling platform of high-quality labeling professionals. The annotator is asked to provide a binary label to indicate whether an image is a true positive of its class. Every image is independently annotated by 3-5 workers to improve the labeling quality, and the final label is reached by majority voting. In total, we have 372,428 annotations over 94,906 images on Mini-ImageNet, and 155,061 annotations over 51,687 images on Stanford Cars, out of which there are 28,691 and 12,639 image with false (noisy) labels. Using these noisy image, we replace $p\%$ of the original training images in Mini-ImageNet and Stanford Cars. Similar to the synthetic noise, $p$ is made uniform across classes, *e.g.* $p = 20\%$ means that every class has roughly 20% false labels. Besides, we also append all annotated images to the original datasets and obtain two larger augmented datasets. The last two rows of Table 1 list the two augmented datasets and their underlying noise levels (19% and 21%). We report their performance in Section 5.3.

Table 1: Overview of the datasets. The same test set, on each dataset, is shared in evaluation.

| Dataset | #Class | Noise Type | Train Size | Test Size | Noise Level(s) |
|---|---|---|---|---|---|
| Red Mini-ImageNet | 100 | image search | 39,000 | 5,000 | 10 (0%-80%) |
| Red Stanford Cars | 196 | image search | 8,144 | 8,041 | 10 (0%-80%) |
| Blue Mini-ImageNet | 100 | uniform flipping | 50,000 | 5,000 | 10 (0%-80%) |
| Blue Stanford Cars | 196 | uniform flipping | 8,144 | 8,041 | 10 (0%-80%) |
| Augmented Mini-ImageNet | 100 | image search | 144,906 | 5,000 | Fixed (19%) |
| Augmented Stanford Cars | 196 | image search | 59,627 | 8,041 | Fixed (21%) |

For comparison, we also construct 10 uniform label-flipping datasets under the same noise levels. For convenience, we will use ***Blue Noise*** to denote the synthetic noise and ***Red Noise*** for the real-world image search noise. Table 1 summarizes the datasets. The test set in each dataset (*i.e.* Mini-ImageNet and Stanford) is shared across all training conditions such that their results are comparable. Red Mini-ImageNet is smaller because we ran out of noisy images for some common classes like "carton" and "hotdog". For common classes, it becomes more difficult to get noisy images. On average, we can get only one noisy label after labeling every 22 images. The size difference in Mini-ImageNet may not be a problem for our study as they have similar test performance and we also verify on Stanford Cars whose size is the same for Blue and Red noise.

---

[4] https://images.google.com/
[5] https://support.google.com/websearch/answer/29508

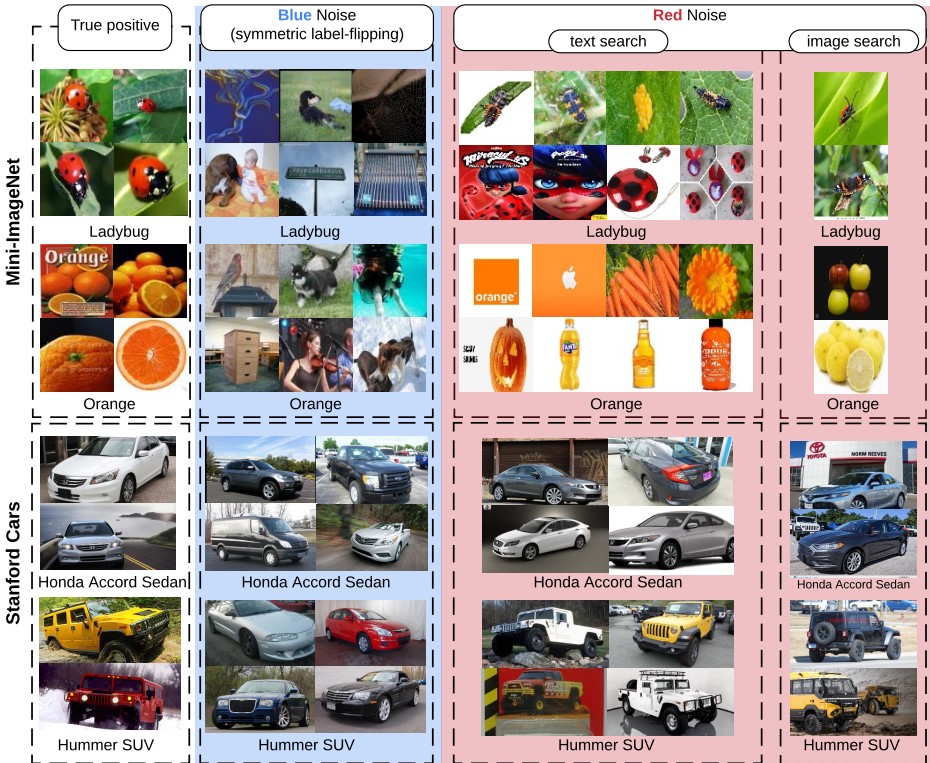

Figure 1: Comparison of uniform label-flipping noise (Blue noise) and image search noise (Red noise). From left to right, columns are true positives, blue noise, red noise (from text-to-image search) and red noise (image-to-image search). The image-to-image search noise (the last column) only accounts for 28% of our data and fewer images are shown as a result.

Fig. 1 illustrates the true positive, blue, and red noisy images. For reference, Fig. 11 in the Appendix shows noisy training images on the WebVision benchmark (Li et al., 2017a) in which the true labels are not provided.

There are two noticeable differences between the blue and red noise:

- Real-world noisy images are more visually or semantically relevant to the true positive images.
- Real-world noisy images may come outside the fixed set of classes in the dataset. For example, the noisy images of "ladybug" include "fly" and other bugs that do not belong to any of the class in Mini-ImageNet.

To understand their differences, we will compare Blue and Red noise in the rest of this paper.

## 4 METHODS

This paper evaluates robust deep learning methods on the introduced benchmark. We select six methods from four directions that deal with noisy training data: (a) regularization, (b) label/prediction correction, (c) example weighting, and (d) vicinal risk minimization. These methods are selected because they (i) represent a reasonable coverage of recent work, (ii) are competitive to the state-of-the-art on the common CIFAR-100 dataset with synthetic noise. Our study seeks the answer to the following questions:

1. How does their performance differ on synthetic versus real-world noise?
2. What is their real performance gap when each method is extensively tuned for every noise level?

As the noise levels span across a wide range from 0% to 80%, we find the hyperparameters of robust DNNs are important. By extensively searching hyperparameters for every noise level, these methods can be very competitive to the state-of-the-art on the commonly used CIFAR-100 with synthetic

noise. See Table 3 in the Appendix. To answer the second question, we need to train a formidable number of experiments, *e.g.* 920 experiments on a single dataset! Daunting as it may seem, the experiments are, however, necessary to ensure the improvement stems from the methodology as opposed to favorable hyperparameter settings.

To briefly introduce these methods, consider a classification problem with training set $\mathcal{D} = \{\mathbf{x}_1, y_1), \cdots, (\mathbf{x}_n, y_n)\}$, where $\mathbf{x}_i$ denotes the $i^{th}$ training image and $y_i \in [1, m]$ is an integer-valued noisy label over $m$ possible classes. Let $g_s(\mathbf{x}_i; \boldsymbol{w})$ denote the prediction of our DNN, parameterized by $\boldsymbol{w} \in \mathbb{R}^d$. In vanilla training, we optimize the following objective:

$$\boldsymbol{w}^* = \min_{\boldsymbol{w} \in \mathbb{R}^d} \frac{1}{n} \sum_{i=1}^{n} \ell(y_i, g_s(\mathbf{x}_i, \boldsymbol{w})) + \theta \|\boldsymbol{w}\|_2^2, \tag{1}$$

where $\ell(y_i, g_s(\mathbf{x}_i, \boldsymbol{w}))$, or $\ell_i$ for short, is the cross-entropy loss with Softmax. $\theta$ is the decay parameter on the $l_2$ norm of the model parameters.

Weight decay and dropout are two classical regularization methods. In ***Weight Decay***: we tune $\theta$ in $\{e^{-5}, e^{-4}, e^{-3}, e^{-2}\}$ and set its default value to $e^{-4}$ which is the best value found on the ILSVRC12 dataset (Deng et al., 2009). For ***Dropout*** (Srivastava et al., 2014), following (Arpit et al., 2017), we apply a large dropout ratio for noisy data and tune its keep probability in $\{0.1, 0.2, 0.3, 0.4, 0.5\}$. By default, we disable dropout in training following the advice in (Kornblith et al., 2019).

We select two methods for label/prediction correction. ***Reed*** (Reed et al., 2014) is a method for correcting the loss with the learned label. The soft version is used for its better performance. Let $\text{softmax}(g_s(\mathbf{x}_i; \boldsymbol{w})) = [q_{i1}, \ldots, q_{im}]$ denote the prediction for the $i^{th}$ image. Reed replaces the loss in Equation 1 with:

$$\ell(y_i, g_s(\mathbf{x}_i, \boldsymbol{w})) = -(\beta \sum_{j=1}^{m} \mathbf{1}_{\text{condition}}(y_i = j) \log(q_{ij}) + (1 - \beta) \sum_{j=1}^{m} q_{ij} \log(q_{ij})), \tag{2}$$

where $\mathbf{1}_{\text{condition}}$ is 1 if the condition is true, 0 otherwise. Reed weights the cross-entropy loss computed over the noisy label (first term) and the learned label (second term). We tune the hyperparameter $\beta$ in $\{0.95, 0.75, 0.5, 0.3\}$.

***S-model*** (Goldberger & Ben-Reuven, 2017) introduces a convenient way to append a new layer to a DNN to learn noise transformation so as to "correct" the predictions. Let $z_i$ denote the unknown true label for the $i^{th}$ image, and $\boldsymbol{q}_i$ and $\hat{\boldsymbol{q}}_i$ denote the original and the learned prediction. It estimates the prediction over the true label by the learned conditional probability:

$$\hat{q}_{ij} = P(z_i = j | \mathbf{x}_i) = \sum_{k=1}^{m} P(z_i = j | y_i = k) P(y_i = k | \mathbf{x}_i) = \sum_{k=1}^{m} P(z_i = j | y_i = k) \frac{\exp\{q_{ik}\}}{\sum_t \exp\{q_{it}\}}. \tag{3}$$

This is implemented as a label transition layer parameterized by $\boldsymbol{B} \in \mathbb{R}^{m \times m}$. We have $\hat{\boldsymbol{q}}_i = \text{softmax}(\boldsymbol{B})\boldsymbol{q}_i$, where the softmax is applied over $\boldsymbol{B}_{i,:}, \forall i \in [1, m]$. According to the paper, we initialize $\boldsymbol{B} = \log((1 - \epsilon)\boldsymbol{I} + \epsilon \times \frac{1}{m-1}\boldsymbol{J})$, where $\boldsymbol{I}$ and $\boldsymbol{J}$ are the identity and the all-one matrix; $\epsilon$ is a small constant set to $e^{-6}$.

***MentorNet*** (Jiang et al., 2018) is a competitive example-weighting method, which aims to assign smaller weights to noisy examples. It introduces the learnable latent weight variable $\boldsymbol{v}$ for every training example and adds a regularization term over the weight variable:

$$\boldsymbol{w}^* = \min_{\boldsymbol{w} \in \mathbb{R}^d, \boldsymbol{v} \in [0,1]^n} \frac{1}{n} \sum_{i=1}^{n} v_i \ell(y_i, g_s(\mathbf{x}_i, \boldsymbol{w})) + \theta \|\boldsymbol{w}\|_2^2 + \sum_{i=1}^{n} \frac{1}{2}\lambda_2 v_i^2 - (\lambda_1 + \lambda_2)v_i, \tag{4}$$

When $\boldsymbol{w}$ is fixed, solving Equation 4 yields a weighting function that is monotonically decreasing with the example loss $\ell_i$:

$$v_i = \begin{cases} \mathbf{1}_{\text{condition}}(\ell_i \leq \lambda_1) & \lambda_2 = 0 \\ \min(\max(0, 1 - \frac{\ell_i - \lambda_1}{\lambda_2}), 1) & \lambda_2 \neq 0 \end{cases}, \tag{5}$$

where $\lambda_1$ and $\lambda_2$ are parameters. We employ the predefined MentorNet to compute the example weight in Equation 5 at the mini-batch level. It tracks the moving average of the $p$-percentile of the loss inside every mini-batch and sets $\lambda_1$ and $\lambda_2$ accordingly. Following the paper, we set the burn-in epoch to 10-20% of the total training epochs and tune the hyperparameter $p$-percentile in $\{85\%, 75\%, 55\%, 35\%\}$.

***Mixup*** (Zhang et al., 2018) is a simple and effective method for robust training. It minimizes the vicinal risk $\ell(\hat{\boldsymbol{y}}_i, g_s(\hat{\mathbf{x}}_i, \boldsymbol{w}))$ calculated from:

$$\hat{\mathbf{x}}_i = \lambda \mathbf{x}_i + (1 - \lambda)\mathbf{x}_j, \qquad \hat{\boldsymbol{y}}_i = \lambda \boldsymbol{y}_i + (1 - \lambda)\boldsymbol{y}_j, \tag{6}$$

where $\boldsymbol{y}_i, \boldsymbol{y}_j \in \mathbb{R}^m$ are two one-hot label vectors and the pairs $(\mathbf{x}_i, \boldsymbol{y}_i)$ and $(\mathbf{x}_j, \boldsymbol{y}_j)$ are drawn at random from the same mini-batch. The mixing weight $\lambda$ is sampled from a conjugate prior Beta distribution $\lambda \sim \mathrm{Beta}(\alpha, \alpha)$ for $\alpha > 0$. Following the paper, we search the hyperparameter $\alpha$ in $\{1, 2, 4, 8\}$ for noisy training data.

For each method, we examine two training settings: (i) fine-tuning from the ImageNet checkpoint and (ii) training from scratch. For method comparison, Inception-ResNet-V2 (Szegedy et al., 2017) is used as the default network architectures. For vanilla training, we also experiment with six other architectures: EfficientNet-B5 (Tan & Le, 2019), MobileNet-V2 (Sandler et al., 2018), ResNet-50 and ResNet-101 (He et al., 2016), Inception-V2 (Ioffe & Szegedy, 2015), and Inception-V3 (Szegedy et al., 2016). The top-1 accuracy of these architectures on the ImageNet ILSVRC 2012 validation ranges from 71.6% to 83.6%. We first train our networks to get the best result on the clean training dataset and fix the setting across all noise levels, *e.g.* learning rate schedule and maximum epochs to train. See Section A.1 in the Appendix for the detailed implementation.

## 5 RESULTS

### 5.1 VANILLA TRAINING

Fig. 2 plots the training curve (gray) and the test curve (colored) on Blue and Red noisy benchmarks using vanilla training. The first two columns of Fig. 2 show the synthetic (in blue) and real-world noise (in red), respectively, where the $x$-axis is the training step. The colored belt plots the 95% confidence interval over 10 noise levels and the solid curve highlights the 40% noise level. The first row in each sub-figure is training from scratch and the second row is fine-tuning. Two classification accuracies on the test set are compared. The *peak accuracy* denotes the maximum test accuracy throughout the training. The *converged accuracy* is the test accuracy after training has converged, which, for most methods, means the training accuracy reaches 100%. See Fig. 2a for an example.

**DNNs generalize much better on the red noise.** By comparing the width of Red and Blue belt in Fig. 2 under the same training condition (row-wise), we can see that the test accuracy's standard deviation is considerably smaller on Red noise than on Blue noise. This indicates a smaller difference in test accuracy between the clean and the noisy training data, suggesting that DNNs generalize better on Red noise.

For a clearer illustration, as an example, we plot networks trained from scratch on Mini-ImageNet in Fig. 3. Specifically, Fig. 3a shows the training accuracy of the 0%, 40%, and 60% noise levels along with the training step. Fig. 3b shows the difference in final converged test accuracies, relative to the accuracy on the clean training data, under 10 noise levels. As the training accuracy is perfect on all noise levels, the drop in the test accuracy can be regarded as an indicator of the generalization gap. The blue curves in Fig 3b confirm Zhang et al. (2017)'s finding that DNNs generalize poorly as synthetic noise levels increase. For example, the test accuracy of EfficientNet will drop by 85% when Blue noise reaches 60%. On the real-world noise, however, the gap is considerably smaller, *e.g.* the drop on 60% Red noise is only 23%. This pattern holds for all architectures in our study. See the curves for EfficientNet, Inception-ResNet, and MobileNet in Fig. 3b.

Results in Fig. 2 and Fig. 3b suggest that DNNs generalize better on real-world noisy data. This phenomenon is probably due to the two properties discussed in Section 3: (i) red noisy images are similar to clean training images, and hence bring less change to the training (ii) red noisy images are often sampled out of the training classes. This may make them less confusing for the fixed training classes.

**DNNs may not learn patterns first on the red noise.** The third column of Fig. 2 illustrates the relative drop between the peak and converged test accuracy, where the $x$-axis is the noise level and the $y$-axis computes the relative difference in percentage, *i.e.* (peak - converged)/peak accuracy. We see that there is almost no drop on the clean data ($x = 0$). The drop starts to grow as the noise level increases. This substantiates Arpit et al. (2017)'s finding that DNNs learn patterns first on noisy data. Early stopping which terminates training at the peak accuracy is thus effective on Blue noise.

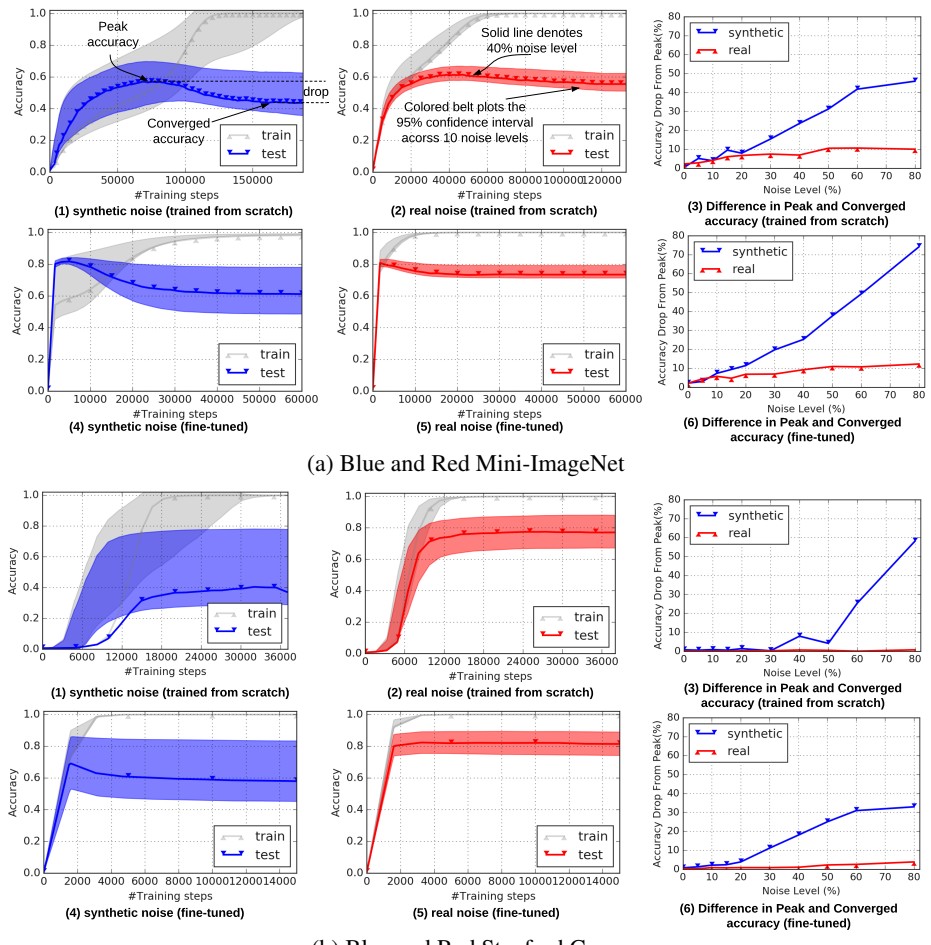

Figure 2: Vanilla training on synthetic (Blue) and real-world noise (Red) using Inception-ResNet.

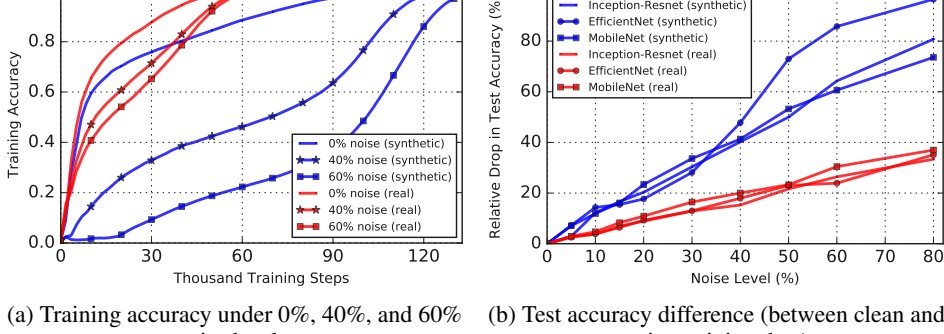

(a) Training accuracy under 0%, 40%, and 60% noise levels.

(b) Test accuracy difference (between clean and noisy training data)

Figure 3: Comparison of generalization on noisy training data of Mini-ImageNet.

However, the drop on Red noise is significantly smaller and even approaches zero on Stanford Cars. See the red curves in the third column of Fig. 2. This suggests DNNs may not learn patterns first on real-world noisy data, especially for the fine-grained classification task. Our hypothesis is that Blue noise images are sampled uniformly from a fixed number of classes, and the uniform errors can be mitigated in the DNN's early training stage before it memorizes all noisy labels. Real-world noisy images are sampled non-uniformly from an infinite number of classes, making it difficult for DNNs to identify meaningful patterns in the red noise.

**ImageNet architectures generalize well on noisy data when the networks are fine-tuned.** Comparing the first and second rows in Fig. 2, we observe that the test accuracy for fine-tuning is higher than that for training from scratch on both Red and Blue noise. This is consistent with (Hendrycks

et al., 2019) where they found pre-training improves model robustness on Blue noise. Our results extend Hendrycks *et al.*'s finding on Red noise. Furthermore, the benefit is more evident on the noisy data. For example, on Stanford Cars, fine-tuning improves training from scratch by less than 1% (91.2% vs. 90.5%) on the clean data. At the 40% noise level, the difference grows to 28.9% (69.3% vs. 40.4%) on Blue noise and 4.8% (82.2% vs. 77.4%) on Red noise. See Table 6 in the Appendix.

In Fig. 4, we compare the fine-tuning performance using ImageNet architectures and compute the correlation coefficient, where the $y$-axis is the peak accuracy and the $x$-axis is the top-1 accuracy of the architecture on ImageNet ILSVRC 2012 validation. The bar plots the 95% confidence interval across 10 noise levels, where the center dot marks the mean. As it shows, there is a decent correlation between the ImageNet accuracy and the test accuracy on noisy data. The Pearson correlation $r$ is 0.897 on Mini-ImageNet and 0.875 on Stanford Cars. The results suggest that, when networks are fine-tuned, ImageNet architectures generalize well on noisy data. This finding generalizes Kornblith et al. (2019)'s recent finding to noisy data.

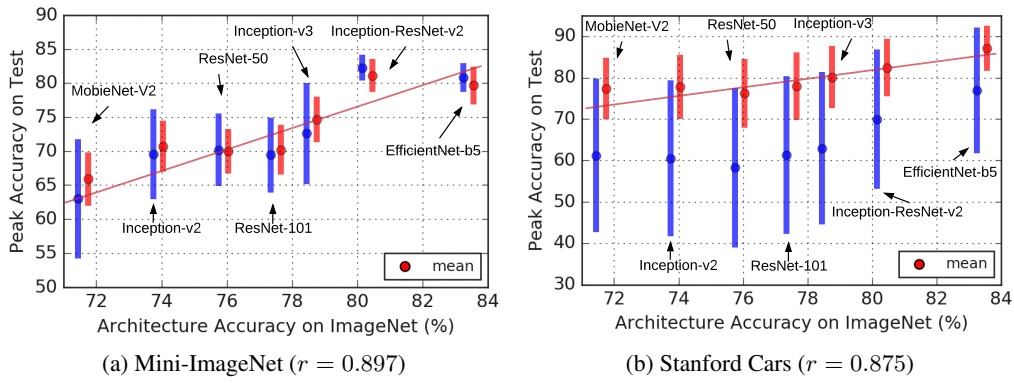

(a) Mini-ImageNet ($r = 0.897$)    (b) Stanford Cars ($r = 0.875$)

Figure 4: Fine-tuning using different ImageNet architectures.

### 5.2 COMPARISON OF THE ROBUST DEEP LEARNING METHODS

In Fig. 5, Fig. 6, Fig. 8, and Fig. 9, we compare the robust deep learning methods on Blue and Red noise, where the $x$-axis shows the peak accuracy and its corresponding 95% confidence interval over different hyperparameters. We mainly compare the peak accuracy which, as shown in recent studies (Song et al., 2018; Arazo et al., 2019), is more challenging to improve. We also list their converged accuracies in the Appendix for reference. First of all, we observe big performance variances (or confidence intervals) in most methods, suggesting that hyperparameters are important for robust learning methods. The best hyperparameter usually varies for different noise levels, and our observation would be very different if the methods were not extensively tuned for each noise level.

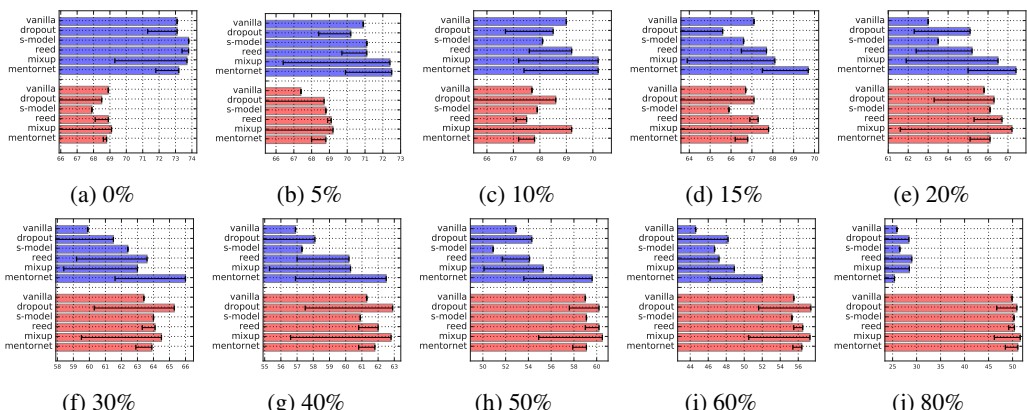

(a) 0%    (b) 5%    (c) 10%    (d) 15%    (e) 20%

(f) 30%    (g) 40%    (h) 50%    (i) 60%    (j) 80%

Figure 5: Peak accuracy of robust DNNs (trained from scratch) on Red and Blue Mini-ImageNet.

**Red noise is more difficult for robust DNNs to improve.** Although robust DNNs are able to improve the performance of vanilla training across all noise levels and types, the improvement is noticeably smaller on Red noise. For example, the average improvement of the best robust method

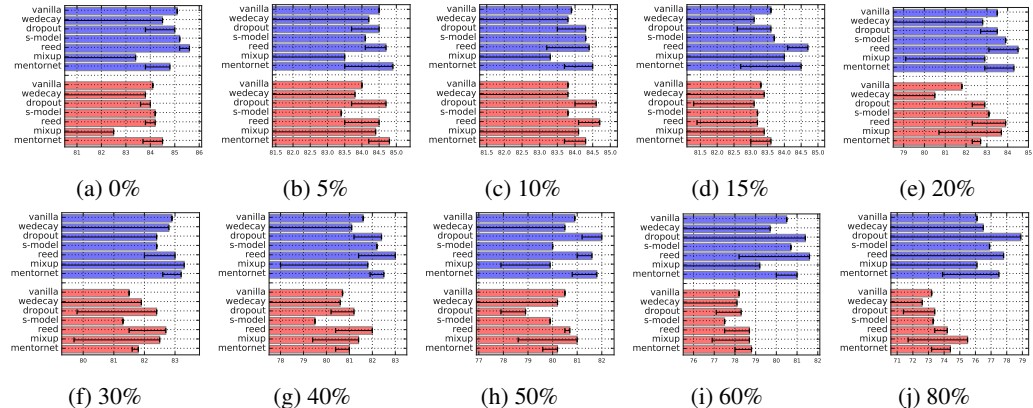

Figure 6: Peak accuracy of robust DNNs (fine-tuned) on Red and Blue Mini-ImageNet.

is 1.25% on Red Mini-ImageNet versus 2.50% on Blue Mini-ImageNet and 4.48% on Red Stanford Cars versus 8.76% on Blue Stanford Cars. The results show that real-world noise is more difficult for robust DNNs to improve.

Comparing methods, we find that no single method performs the best across all noise levels and types. Methods that work well on synthetic noise may not work as well on real-world noise, and vice versa. To be specific, Dropout is effective for training from scratch on Stanford Cars and achieves the best accuracy in 20 trials. Weight Decay mainly benefits fine-tuning but only to a small extent (6 best trials). Reed achieves the best result in 10 trials, all of which are fine-tuning on Mini-ImageNet. S-Model yields marginal gains over the vanilla training. Finally, MentorNet and Mixup achieve the best accuracy in 21 and 23 trials, respectively. Unlike MentorNet, Mixup seems to be more effective on Red noise, suggesting that pair-wise image mixing is more effective than example weighting on real-world noise.

## 5.3 AUGMENTING DATASET WITH NOISY EXAMPLES

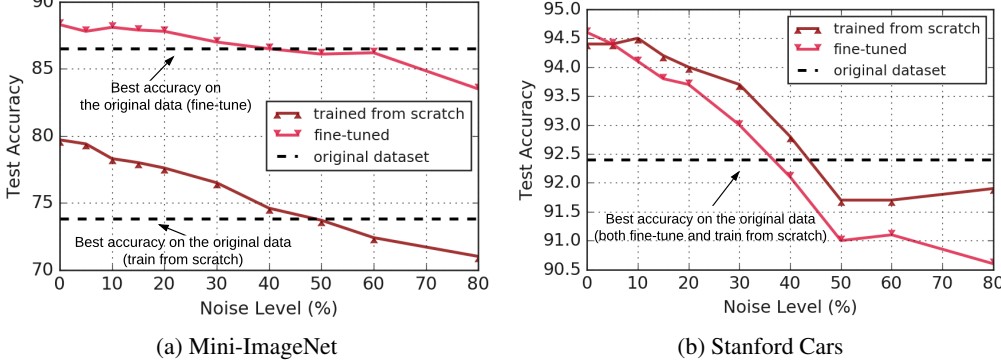

(a) Mini-ImageNet

(b) Stanford Cars

Figure 7: Peak accuracy for different levels of added training noise.

In practice, a simple technique to improve performance is to add noisy or weakly-labeled examples to an existing clean dataset. It is convenient because noisy examples can be automatically collected without any manual labeling effort. Multiple studies have shown that noisy examples can be beneficial for training *e.g.* (Krause et al., 2016; Liang et al., 2016; Mahajan et al., 2018). In particular, Guo et al. (2018) found that adding WebVision noisy images to clean ImageNet can lead to a performance gain. On the other hand, it may be disadvantageous if the true labels of all added examples are incorrect. An interesting question that has yet been answered is *what is the maximum noise level at which the added noisy data can be useful?*.

Our benchmark allows for investigating this question using real-world noisy data in a controlled setting. To do so, we add ∼30K and ∼25K additional images to the original training sets of Mini-ImageNet and Stanford Cars, respectively. The sizes are selected to be 60% and 300% of the original datasets to examine different settings. We control the noise level of the added images in 10 different levels from 0% to 80%. Fig. 7 shows the peak test accuracy (the $y$-axis) across ten noise levels

(the $x$-axis). The dashed line represents the best accuracy obtained on the original dataset. As it shows, the test accuracy generally decreases as the noise level grows. Small noise is useful but large noise can hurt the performance. The equilibrium occurs between 30% to 50%. In all cases, it is useful if the noise level is under 30%. When the noise level is below 30%, it also improves the full augmented datasets in Table 1, the accuracies of which are 0.770 (trained from scratch) and 0.865 (fine-tuned) on Mini-ImageNet, and 0.927 and 0.932 on Stanford Cars. Note that the 30% threshold just represents the observation on our benchmark and should not be overinterpreted.

## 6 CONCLUSION

In this paper, we established a benchmark for controlled real-world noise. On the benchmark, we conducted a large-scale study to understand deep learning on noisy data across a variety of settings. Our studies revealed a number of new findings, improving our understanding of deep learning on noisy data. By comparing six robust deep learning methods, we found that real-world noise is more difficult to improve and methods that work well on synthetic noise may not work as well on real-world noise, and vice versa. This encourages future research to be also carried out on controlled real-world noise. We hope our benchmark, as well as our findings, will facilitate deep learning research on real-world noisy data.

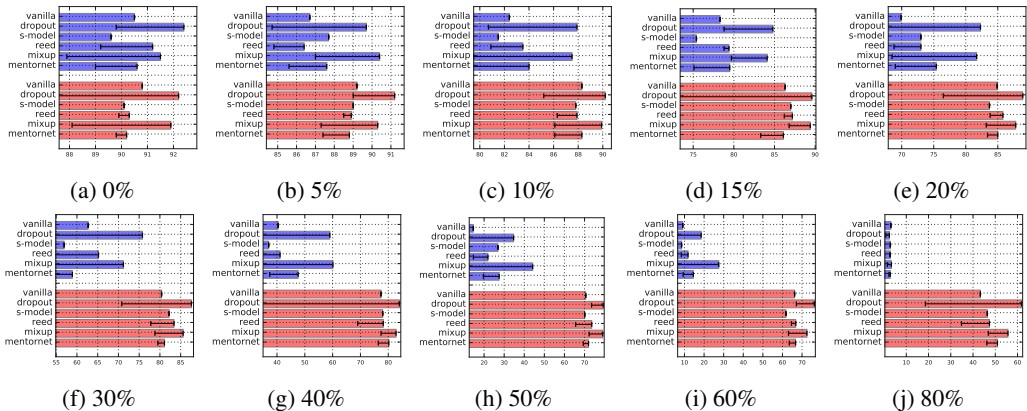

Figure 8: Peak accuracy of robust DNNs (trained from scratch) on Red and Blue Stanford Cars.

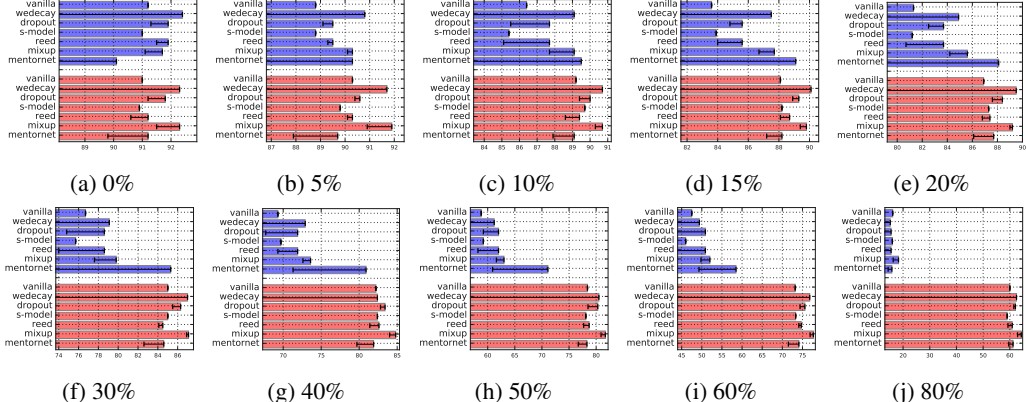

Figure 9: Peak accuracy of robust DNNs (fine-tuned) on Red and Blue Stanford Cars.

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

# A APPENDIX

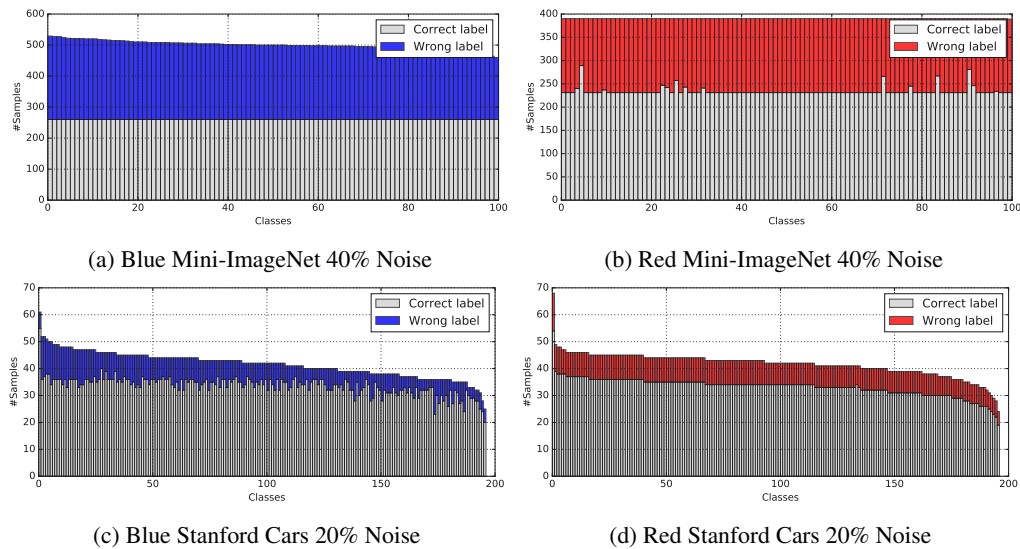

(a) Blue Mini-ImageNet 40% Noise

(b) Red Mini-ImageNet 40% Noise

(c) Blue Stanford Cars 20% Noise

(d) Red Stanford Cars 20% Noise

Figure 10: The distribution of noisy and clean images in Mini-ImageNet and Stanford Cars. Grey, Blue and, Red bar represent images of clean labels, synthetic noisy labels and image-search noisy labels, respectively. Classes are ranked by the number of training examples. Better viewed in color.

## A.1 IMPLEMENTATION DETAILS

This subsection presents the implementation details.

**Architectures:** Table 2 lists the parameter count and input image size for each network architecture examined. We obtained their model checkpoints trained on the ImageNet 2012 dataset from TensorFlow Slim [6], EfficienNet TPU[7], and from the authors of (Kornblith et al., 2019). The last two columns list the top-1 accuracy of our obtained models along with the accuracy reported in the original paper. As shown, the top-1 accuracy of these architectures on the ImageNet ILSVRC 2012 validation ranges from 71.6% to 83.6%, making them suitable for our correlation study.

---

[6]https://github.com/tensorflow/models/tree/master/research/slim

[7]https://github.com/tensorflow/tpu/tree/master/models/official/efficientnet

Table 2: Overview of the ImageNet architectures used in our study.

| Network | Parameters | Image Size | ImageNet Top-1 Acc. | |
|---|---|---|---|---|
| | | | Paper | Our checkpoint |
| EfficientNet B5 (Tan & Le, 2019) | 28.3M | 456 | 83.3 | 83.3 |
| Inception V2 (Ioffe & Szegedy, 2015) | 10.2M | 224 | 74.8 | 73.9 |
| Inception V3 (Szegedy et al., 2016) | 21.8M | 299 | 78.8 | 78.6 |
| Inception-ResNet V2 (Szegedy et al., 2017) | 54.2M | 299 | 80.0 | 80.3 |
| MobileNet V2 (Sandler et al., 2018) | 2.2M | 224 | 72.0 | 71.6 |
| ResNet 50 V1 (He et al., 2016) | 23.5M | 224 | 75.2 | 75.9 |
| ResNet 101 V1 (He et al., 2016) | 42.5M | 224 | 76.4 | 77.5 |

**Training from scratch (random initialization)**: for vanilla training, we trained each architecture on the clean dataset (0% noise level) to find the optimal training setting by grid search. Our grid search consisted of 6 start learning rates of $\{1.6, 0.16, 1.0, 0.5, 0.1, 0.01\}$ and 3 learning rate decay epochs of $\{1, 2, 3\}$. The exponential learning rate decay factor was fixed to 0.975. We trained each network to full convergence to ensure its training accuracy reached 1.0. The maximum epoch to train was 200 on Mini-ImageNet (Red and Blue) and 300 epochs on Stanford Cars (Blue and Red), where the learning rate warmup (Goyal et al., 2017) was used in the first 5 epochs. The training was using Nesterov momentum with a momentum parameter of 0.9 at a batch size of 64, taking an exponential moving average of the weights with a decay factor of 0.9999. We had to reduce the batch size to 8 for EfficientNet for its larger image input. Following Kornblith et al. (2019), our vanilla training was with batch normalization layers but without label smoothing, dropout, or auxiliary heads. We employed the standard prepossessing in EfficientNet[8] for data augmentation and evaluated on the central cropped images on the test set. Training in this way, we obtained reasonable performance on the clean Stanford Cars test set. For example, our Inception-ResNet-V2 got 90.8 (without dropout) and 92.4 (with dropout) versus 89.9 reported in (Kornblith et al., 2019).

**Fine-tuning from ImageNet checkpoint:** for fine-tuning experiments, we initialized networks with ImageNet-pretrained weights. We used a similar training protocol for fine-tuning as training from scratch. The start learning rate was stable in fine-tuning so we fixed it to 0.01 and only searched the learning rate decay epochs in $\{1, 3, 5, 10\}$. Learning rate warmup was not used in fine-tuning. As fine-tuning converges faster, we scaled down the maximum number of epoch to train by a factor of 2. In this case, the training accuracy was still able to reach 1.0. Training in this way, we obtained reasonable performance on the clean Stanford Cars test set. For example, our Inception-ResNet-V2 got 92.4 versus 92.0 reported in (Kornblith et al., 2019) and our EfficientNet-B5 got 93.8% versus 93.6% reported in (Tan & Le, 2019).

**Robust deep learning method comparison:** For method comparison, we used Inception-ResNet as the default network. We fixed the optimal setting found on the clean training set for all methods and all noise levels. We found the hyperparameter for robust DNNs is important. Therefore, we extensively searched the hyperparameter of each method for every noise level and every noise type, using the range discussed in the main manuscript. Comparing 6 methods using all hyperparameters, 10 noise levels, 2 noise types, and 2 training conditions led to a total of 1,840 experiments on two datasets. The mean and 95% confidence interval over different hyperparameters were shown in Fig. 5, Fig. 6, Fig. 8, and Fig. 9 in the main manuscript. The best peak accuracy (along with the converged accuracy) could be found in Table 4 to Table 7. For Dropout, as it converges slower, we added another 100 epochs to its maximum epochs to train.

## A.2 COMPARISON TO THE STATE-OF-THE-ART ON CIFAR-100

This subsection shows that our examined robust learning methods are able to achieve the performance that is comparable to, or even better than, the state-of-the-art on the commonly used synthetic noisy dataset in the literature. Following (Jiang et al., 2018; Shu et al., 2019; Arazo et al., 2019), the noisy CIFAR-100 data are of uniform label-flipping noise, where the label of each image is independently changed to a uniform (incorrect) class with probability $p$, where $p$ is the noise level and is set to 20%, 40%, 60%, and 80%. The clean test images on CIFAR-100 are used for evaluation.

---

[8]https://github.com/tensorflow/tpu/blob/master/models/official/efficientnet/preprocessing.py

Table 3 shows the results where † marks our implementation of MentorNet (Jiang et al., 2018)[9] and Mixup (Zhang et al., 2018)[10] under the best hyperparameter setting[11]. First, the comparison between Row 2 and Row 7 in the table shows extensive hyperparameter search leads to about a 2% gain over the published results in (Jiang et al., 2018). Second, comparing Row 5 and 6 to Row 4, it shows that our examined methods are comparable to the state-of-the-art except for the 40% noise level. Finally, comparing Row 7 and 8 to others, we find that our examined methods are able to achieve the best result on this dataset.

Table 3: The peak test accuracy (%) on CIFAR-100 with synthetic noise. † marks our implementation of MentorNet and Mixup under the best hyperparameter setting. "-" indicates the number is unavailable in the published paper. The best accuracy is in bold.

| Row | Method | Network | Noise level (%) | | | |
|-----|--------|---------|----|----|----|----|
| | | | 20 | 40 | 60 | 80 |
| 1 | Forward (Patrini et al., 2017) | ResNet-44 | 64.0 | - | - | |
| 2 | MentorNet (Jiang et al., 2018) | ResNet-101 | 73.0 | 68.0 | - | 35.0 |
| 3 | Meta-Weight(Shu et al., 2019) | ResNet-28 | - | 67.3 | 58.7 | - |
| 4 | MD-DYR-SH (Arazo et al., 2019) | ResNet-18 | 73.7 | 70.1 | 59.5 | 39.5 |
| 5 | Our MentorNet (Jiang et al., 2018) † | ResNet-18 | 73.5 | 68.5 | **61.2** | 32.5 |
| 6 | Our Mixup (Zhang et al., 2018) † | ResNet-18 | 73.9 | 66.8 | 58.8 | 40.1 |
| 7 | Our MentorNet (Jiang et al., 2018) † | ResNet-101 | 74.2 | **70.9** | 59.5 | 37.8 |
| 8 | Our Mixup (Zhang et al., 2018)† | ResNet-101 | **75.4** | 68.9 | 60.0 | **40.8** |

Noisy Training Examples in the Webvision Dataset

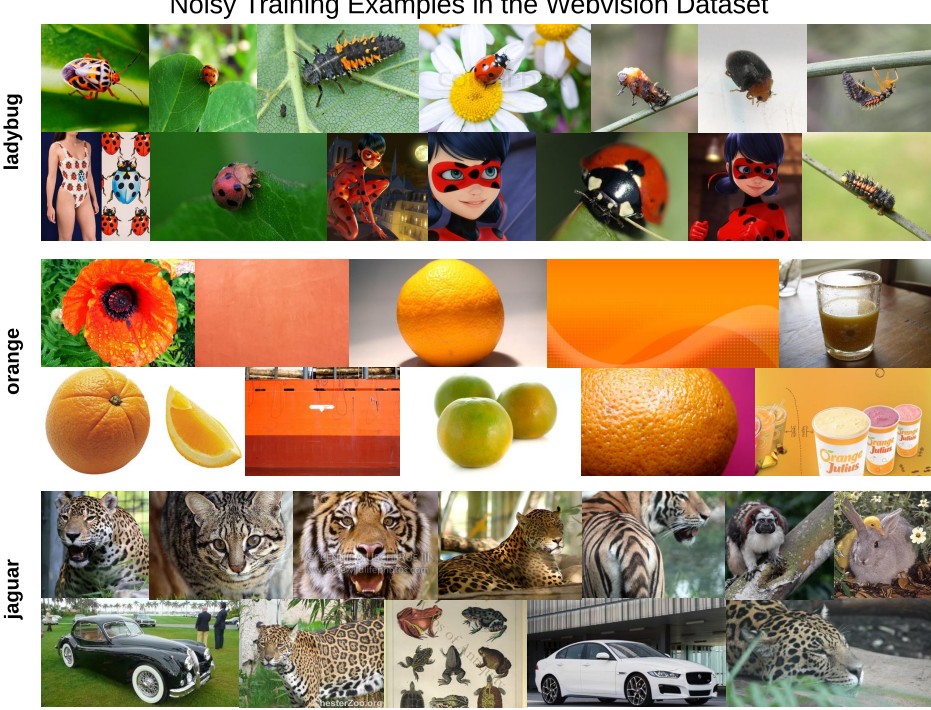

Figure 11: Example training images in the WebVision dataset on three classes: ladybug, orange and jaguar. Note the ground-truth labels are not provided in WebVision and the images above may be true or false positive. For comparison, Fig. 1 shows noisy training images in the proposed dataset which are all manually labeled.

---

[9]Adapted from https://github.com/google/mentornet

[10]Adapted from https://github.com/hongyi-zhang/mixup

[11]The performance would be 2-6% worse If we did not search their hyper-parameters for every noise level.

## A.3 TEXT-TO-IMAGE SEARCH ONLY RESULTS

In this section, we exclude the images brought by image-to-image search and reconduct our main experiments. This data subset contains only the images from Google text-to-image search and hence is similar to the noisy datasets used in previous studies (Li et al., 2017a; Krause et al., 2016; Chen & Gupta, 2015) (note that existing datasets do not have controlled noise). Our goal is to verify whether our findings still hold on this new data subset.

We use *dark red* to denote the text-to-image search only noise and compare it with the red noise reported in the paper. To be specific, the vanilla training and test curves are compared in Fig. 12. The generalization errors are compared in Fig. 13. The finetuning models on different ImageNet architectures are compared in Fig. 14.

As we see in all cases, text-to-image only noise (dark red) performs very similarly to red noise studied in the paper. We can confirm the our findings still hold. That is (1) DNNs generalize much better on the dark red noise (Fig. 12 and Fig. 13). (2) DNNs may not learn patterns first on the dark red noise (Fig. 12). (3) ImageNet architectures generalize well on noisy data when the networks are fine-tuned (Fig. 14). The above results show that the red noise studied in our paper is consistent with the text-to-image only noise. As red noise contains a more diverse types of label noises, we keep it as our main datasets.

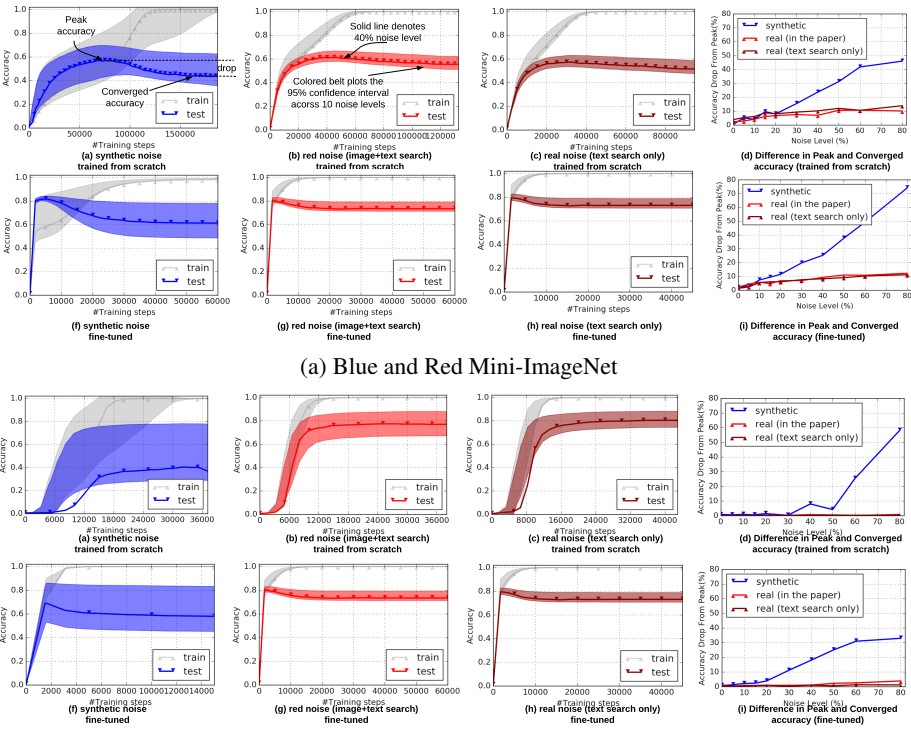

Figure 12: Vanilla training on blue noise and red noise using Inception-ResNet. The first two columns are copied from Fig. 2 for comparison. The dark red curve in the third and fourth column represents the noise of text-to-image search only and can be used to compare with the red curve.

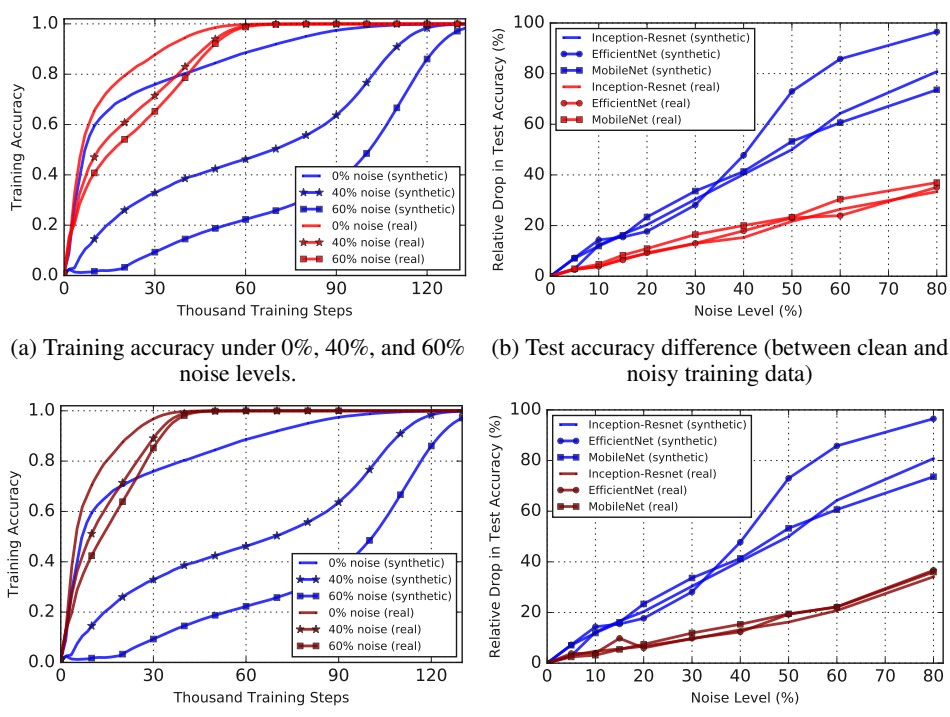

(a) Training accuracy under 0%, 40%, and 60% noise levels.

(b) Test accuracy difference (between clean and noisy training data)

(c) Training accuracy under 0%, 40%, and 60% noise levels (text-to-image search only).

(d) Test accuracy difference (text-to-image search only)

Figure 13: Comparison of generalization on the noisy training data of Mini-ImageNet. The top row is copied from Fig. 3 for comparison. The dark red curve in the bottom row represents the noise of text-to-image search only.

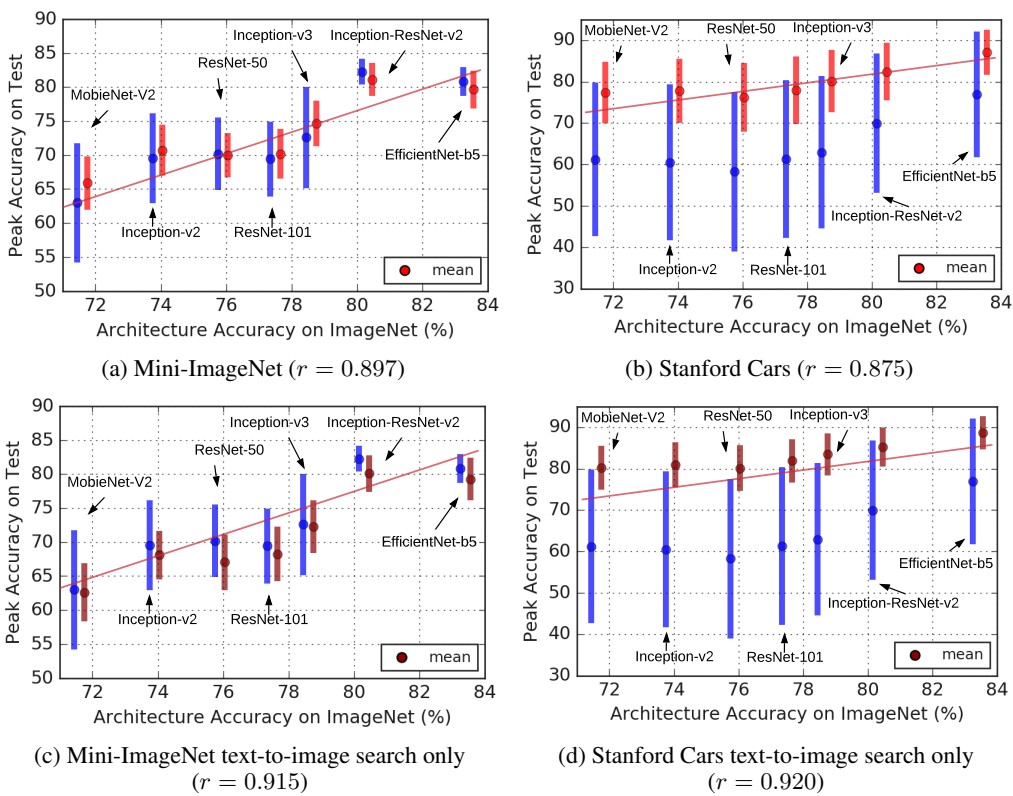

(a) Mini-ImageNet ($r = 0.897$)

(b) Stanford Cars ($r = 0.875$)

(c) Mini-ImageNet text-to-image search only
($r = 0.915$)

(d) Stanford Cars text-to-image search only
($r = 0.920$)

Figure 14: Fine-tuning using different ImageNet architectures. The top row is copied from Fig. 4 for comparison. In the bottom row, the bars in dark red represent the noise of text-to-image search only. The results are comparable to the ones in Fig. 4 with slightly higher correlations (from 0.897 and 0.875 to 0.915 and 0.920).

## A.4 Numerical Results in Method Comparison

This subsection presents the numerical results for the best trial in Fig. 5, Fig. 6, Fig. 8, and Fig. 9 in the main manuscript.

Table 4: Peak accuracy (%) of the best trial for each method, fine-tuned on Mini-ImageNet. The peak and converged test accuracies are shown in the format of XXX/YYY.

| Type | Noise Level | Vanilla | WeDecay | Dropout | S-Model | Reed Soft | Mixup | MentorNet |
|------|-------------|---------|---------|---------|---------|-----------|-------|-----------|
| Blue Noise | 0 | 85.1/83.3 | 84.5/81.9 | 85.0/83.6 | 85.2/83.5 | **85.6**/84.3 | 83.4/82.0 | 84.8/83.4 |
| | 5 | 84.5/82.0 | 84.2/79.7 | 84.5/80.8 | 84.1/81.5 | 84.7/81.9 | 83.5/82.0 | **84.9**/84.5 |
| | 10 | 83.9/77.8 | 83.8/75.9 | 84.3/78.0 | 84.3/78.2 | 84.4/81.5 | 83.3/80.1 | **84.5**/84.3 |
| | 15 | 83.6/75.8 | 83.1/73.8 | 83.6/74.7 | 83.7/76.1 | **84.7**/80.4 | 84.0/79.5 | 84.5/81.2 |
| | 20 | 83.5/74.0 | 82.8/71.9 | 83.5/73.9 | 83.9/73.7 | **84.5**/79.2 | 82.9/77.9 | 84.3/76.5 |
| | 30 | 82.9/66.6 | 82.8/64.1 | 82.4/67.5 | 82.4/66.0 | 83.0/76.7 | **83.3**/74.6 | 83.2/74.0 |
| | 40 | 81.6/61.0 | 81.1/58.5 | 82.4/59.0 | 82.2/60.2 | **83.0**/78.3 | 81.8/70.8 | 82.5/80.4 |
| | 50 | 80.9/50.5 | 80.5/47.9 | **82.0**/68.1 | 80.0/50.8 | 81.6/53.1 | 79.9/64.6 | 81.8/71.0 |
| | 60 | 80.5/40.9 | 79.7/40.3 | 81.4/38.9 | 80.7/42.3 | **81.6**/45.2 | 79.2/63.6 | 81.0/79.8 |
| | 80 | 76.1/19.7 | 76.5/17.6 | **78.9**/24.7 | 76.9/20.2 | 77.8/22.2 | 76.1/60.0 | 77.5/32.3 |
| Red Noise | 0 | 84.1/82.3 | 83.8/81.6 | 84.0/82.7 | 84.2/82.3 | 84.2/82.2 | 82.5/81.9 | **84.5**/84.0 |
| | 5 | 84.0/80.4 | 83.8/80.4 | 84.7/80.8 | 83.4/80.6 | 84.5/81.1 | 84.4/82.5 | **84.8**/84.5 |
| | 10 | 83.8/79.0 | 83.8/78.3 | 84.6/79.1 | 83.8/78.7 | **84.7**/79.9 | 84.1/81.2 | 84.3/83.4 |
| | 15 | 83.3/79.4 | 83.4/78.0 | 83.1/78.8 | 83.2/79.1 | 83.2/78.8 | 83.4/80.7 | **83.6**/82.5 |
| | 20 | 81.8/76.2 | 80.5/76.3 | 82.9/77.5 | 83.1/77.9 | **83.9**/77.5 | 83.7/80.3 | 82.7/82.4 |
| | 30 | 81.5/75.9 | 81.9/75.1 | 82.4/74.2 | 81.3/75.6 | **82.7**/75.8 | 82.5/79.2 | 81.8/81.7 |
| | 40 | 80.7/73.3 | 80.6/72.7 | 81.2/71.7 | 79.5/73.8 | **82.0**/73.8 | 81.4/76.8 | 81.0/80.8 |
| | 50 | 80.5/71.7 | 80.2/71.1 | 78.9/70.3 | 79.9/72.1 | 80.7/72.9 | **81.0**/76.3 | 80.2/78.5 |
| | 60 | 78.2/69.8 | 78.1/67.8 | 78.3/66.5 | 77.5/68.8 | 78.7/69.5 | 78.7/75.9 | **78.8**/76.4 |
| | 80 | 73.2/64.3 | 72.6/63.9 | 73.4/63.9 | 73.3/65.4 | 74.2/63.7 | **75.5**/70.0 | 74.4/69.2 |

Table 5: Peak accuracy (%) of the best trial of for method, trained from scratch on Mini-ImageNet. '-' denotes the method that is failed to converge.

| Type | Noise Level | Vanilla | WeDecay | Dropout | S-Model | Reed Soft | Mixup | MentorNet |
|------|-------------|---------|---------|---------|---------|-----------|-------|-----------|
| Blue Noise | 0 | 73.1/72.8 | -/- | 73.1/67.9 | 73.8/71.8 | **73.8**/71.2 | 73.7/73.0 | 73.2/71.5 |
| | 5 | 70.9/70.7 | -/- | 70.2/63.4 | 71.1/67.1 | 71.1/66.7 | 72.4/70.3 | **72.5**/69.5 |
| | 10 | 69.0/63.9 | -/- | 68.5/60.5 | 68.1/63.6 | 69.2/63.4 | 70.2/67.2 | **70.2**/67.9 |
| | 15 | 67.1/60.7 | -/- | 65.6/56.7 | 66.6/61.1 | 67.7/60.5 | 68.1/63.2 | **69.7**/66.1 |
| | 20 | 63.0/58.0 | -/- | 65.1/53.4 | 63.5/57.7 | 65.2/57.6 | 66.5/60.6 | **67.4**/65.8 |
| | 30 | 59.9/50.7 | -/- | 61.5/47.2 | 62.4/50.8 | 63.6/52.2 | 63.0/54.6 | **66.0**/64.2 |
| | 40 | 56.9/43.5 | -/- | 58.1/40.2 | 57.3/45.0 | 60.2/44.5 | 60.3/46.9 | **62.5**/62.1 |
| | 50 | 52.9/36.4 | -/- | 54.3/33.5 | 50.9/36.1 | 54.1/36.8 | 55.3/42.8 | **59.6**/57.7 |
| | 60 | 44.6/26.0 | -/- | 48.2/23.3 | 46.7/27.1 | 47.2/27.2 | 48.9/47.4 | **52.0**/47.2 |
| | 80 | 25.9/14.0 | -/- | 28.4/18.8 | 26.5/18.7 | **29.0**/12.7 | 28.5/28.5 | 25.4/18.5 |
| Red Noise | 0 | 68.9/67.4 | -/- | 68.5/63.9 | 67.9/65.5 | 68.9/65.8 | **69.1**/68.6 | 68.8/65.8 |
| | 5 | 67.4/65.5 | -/- | 68.7/61.6 | 68.8/65.7 | 69.1/64.8 | **69.2**/67.4 | 68.8/65.1 |
| | 10 | 67.7/64.9 | -/- | 68.6/60.2 | 67.9/62.6 | 67.5/63.5 | **69.2**/66.7 | 67.8/63.5 |
| | 15 | 66.7/62.7 | -/- | 67.1/58.7 | 65.9/60.7 | 67.3/65.5 | **67.8**/64.8 | 66.8/62.7 |
| | 20 | 65.8/61.4 | -/- | 66.3/56.7 | 66.1/59.9 | 66.7/60.5 | **67.2**/63.6 | 66.1/62.2 |
| | 30 | 63.4/58.7 | -/- | **65.3**/54.3 | 64.0/56.4 | 64.1/57.0 | 64.5/61.3 | 63.9/59.8 |
| | 40 | 61.3/57.1 | -/- | **62.9**/51.5 | 60.9/54.1 | 62.0/53.0 | 62.8/58.7 | 61.8/57.7 |
| | 50 | 59.0/52.8 | -/- | 60.2/47.9 | 59.1/52.7 | 60.2/51.6 | **60.5**/55.1 | 59.1/56.0 |
| | 60 | 55.5/49.6 | -/- | **57.4**/45.6 | 55.3/47.6 | 56.5/47.9 | 57.3/52.5 | 56.4/51.6 |
| | 80 | 49.9/44.9 | -/- | 50.9/39.3 | 50.3/42.7 | 50.4/42.8 | **51.6**/47.4 | 51.1/46.7 |

Table 6: Peak accuracy (%) of the best trial for each method, fine-tuned on Stanford Cars. The peak and converged test accuracies are shown in the format of XXX/YYY.

| Type | Noise Level | Vanilla | WeDecay | Dropout | S-Model | Reed Soft | Mixup | MentorNet |
|---|---|---|---|---|---|---|---|---|
| Blue Noise | 0 | 91.2/90.6 | **92.4**/92.2 | 91.9/91.3 | 91.0/90.7 | 91.3/91.0 | 91.7/91.6 | 90.1/90.0 |
| | 5 | 88.8/87.7 | **90.8**/90.5 | 89.5/88.4 | 88.8/88.3 | 88.8/88.8 | 90.3/90.0 | 90.3/89.8 |
| | 10 | 86.4/84.6 | 89.1/87.9 | 87.7/85.1 | 85.4/84.2 | 85.4/87.5 | 89.1/88.9 | **89.5**/89.5 |
| | 15 | 83.6/81.7 | 87.5/86.4 | 85.6/81.7 | 83.9/81.4 | 83.9/86.4 | 87.7/87.2 | **89.1**/89.1 |
| | 20 | 81.3/78.2 | 84.9/82.9 | 83.7/77.4 | 81.2/78.1 | 81.2/83.9 | 85.6/85.6 | **88.1**/87.8 |
| | 30 | 76.7/68.2 | 79.1/75.3 | 78.6/67.0 | 75.7/68.5 | 75.7/79.8 | 79.8/76.4 | **85.3**/85.2 |
| | 40 | 69.3/56.8 | 72.9/63.8 | 71.9/56.0 | 69.7/58.2 | 69.7/71.8 | 73.6/68.1 | **80.9**/79.3 |
| | 50 | 58.8/44.1 | 61.2/48.7 | 62.0/44.2 | 59.2/45.1 | 59.2/60.0 | 63.0/56.2 | **71.1**/66.7 |
| | 60 | 47.5/32.8 | 49.4/36.9 | 50.9/31.8 | 46.0/32.4 | 46.0/47.8 | 52.0/43.6 | **58.5**/57.0 |
| | 80 | 16.1/10.8 | 15.2/10.0 | 15.5/10.1 | 16.0/10.6 | 16.0/15.9 | **18.3**/17.2 | 15.8/13.8 |
| Red Noise | 0 | 91.0/90.7 | **92.3**/92.1 | 91.8/91.2 | 90.9/90.7 | 91.2/90.8 | 92.3/92.3 | 91.2/91.1 |
| | 5 | 90.3/90.1 | 91.7/91.7 | 90.6/89.6 | 89.8/89.2 | 90.3/89.6 | **91.9**/91.8 | 89.7/89.3 |
| | 10 | 89.2/88.5 | **90.7**/90.5 | 90.0/89.1 | 89.7/89.1 | 89.4/88.9 | 90.7/90.5 | 89.1/88.7 |
| | 15 | 88.1/87.5 | **90.1**/89.5 | 89.3/88.1 | 88.2/87.8 | 88.7/88.4 | 89.8/89.7 | 88.2/87.8 |
| | 20 | 86.9/86.2 | **89.5**/89.0 | 88.4/87.0 | 87.3/86.8 | 87.4/86.1 | 89.2/89.0 | 87.7/86.7 |
| | 30 | 85.0/84.3 | 87.0/86.0 | 86.3/84.0 | 85.0/84.2 | 84.5/83.8 | **87.1**/86.9 | 84.6/84.3 |
| | 40 | 82.2/81.4 | 82.4/81.3 | 83.4/81.7 | 82.4/80.9 | 82.6/81.6 | **84.8**/84.3 | 81.9/81.0 |
| | 50 | 78.4/76.7 | 80.5/80.0 | 80.3/77.2 | 78.1/76.6 | 78.7/76.7 | **81.7**/81.6 | 78.3/76.8 |
| | 60 | 73.2/71.4 | 76.8/75.0 | 75.6/72.1 | 73.3/71.6 | 74.7/72.1 | **77.7**/77.4 | 74.1/72.9 |
| | 80 | 60.0/57.7 | 62.5/61.2 | 62.1/57.3 | 59.0/56.8 | 60.9/58.1 | **64.3**/63.0 | 61.2/57.2 |

Table 7: Peak accuracy (%) of the best trial for each method, trained from scratch on Stanford Cars. '-' denotes the method that is failed to converge.

| Type | Noise Level | Vanilla | WeDecay | Dropout | S-Model | Reed Soft | Mixup | MentorNet |
|---|---|---|---|---|---|---|---|---|
| Blue Noise | 0 | 90.5/89.9 | -/- | **92.4**/92.4 | 89.6/89.5 | 91.2/91.2 | 91.5/91.3 | 90.6/90.4 |
| | 5 | 86.7/86.3 | -/- | 89.7/89.6 | 87.7/87.6 | 86.4/86.2 | **90.4**/90.4 | 87.6/87.6 |
| | 10 | 82.4/81.7 | -/- | **87.9**/87.6 | 81.5/81.4 | 83.5/83.1 | 87.5/87.4 | 84.0/83.9 |
| | 15 | 78.3/77.9 | -/- | **84.8**/84.5 | 75.4/75.2 | 79.4/79.1 | 84.1/84.0 | 79.5/79.3 |
| | 20 | 69.9/69.0 | -/- | **82.3**/82.2 | 73.0/72.8 | 73.0/72.7 | 81.7/81.6 | 75.4/75.2 |
| | 30 | 62.7/62.6 | -/- | **75.8**/71.1 | 56.9/56.4 | 65.2/64.7 | 71.2/71.1 | 58.9/57.7 |
| | 40 | 40.4/37.2 | -/- | 59.0/58.7 | 37.0/37.0 | 41.1/40.7 | **60.1**/60.0 | 47.6/47.5 |
| | 50 | 14.6/14.0 | -/- | 34.7/33.2 | 26.9/26.6 | 21.9/21.9 | **44.2**/42.9 | 27.5/27.5 |
| | 60 | 9.1/6.8 | -/- | 18.5/18.0 | 8.5/8.5 | 11.8/11.3 | **27.5**/25.4 | 14.5/14.2 |
| | 80 | 3.1/1.3 | -/- | 2.4/2.3 | 2.8/2.8 | 2.8/2.7 | **3.3**/3.0 | 2.8/2.7 |
| Red Noise | 0 | 90.8/90.8 | -/- | **92.2**/92.2 | 90.1/90.1 | 90.3/90.0 | 91.9/91.9 | 90.2/90.1 |
| | 5 | 89.2/89.2 | -/- | **91.2**/90.8 | 89.0/88.9 | 88.9/88.8 | 90.3/90.2 | 88.8/88.6 |
| | 10 | 88.3/88.3 | -/- | **90.2**/90.2 | 87.8/87.8 | 87.9/87.7 | 89.9/89.9 | 88.3/88.3 |
| | 15 | 86.3/86.3 | -/- | **89.6**/89.6 | 87.0/86.9 | 87.2/87.2 | 89.4/89.1 | 86.1/59.9 |
| | 20 | 84.9/84.7 | -/- | **88.9**/88.9 | 83.7/83.6 | 85.8/85.7 | 87.8/87.6 | 85.0/84.8 |
| | 30 | 80.4/80.2 | -/- | **87.6**/87.6 | 82.2/81.9 | 83.4/83.0 | 85.6/85.2 | 81.1/80.9 |
| | 40 | 77.4/76.9 | -/- | **84.0**/84.0 | 78.0/77.8 | 78.2/77.8 | 82.8/82.5 | 80.2/76.9 |
| | 50 | 70.6/70.3 | -/- | **79.3**/79.2 | 70.1/70.1 | 73.6/73.5 | 79.1/78.9 | 72.0/72.0 |
| | 60 | 66.2/66.2 | -/- | **76.3**/75.9 | 61.8/61.4 | 66.8/66.6 | 72.5/72.1 | 66.7/66.6 |
| | 80 | 43.3/43.0 | -/- | **61.8**/61.8 | 46.4/46.4 | 47.4/46.7 | 55.7/55.4 | 51.0/50.9 |

