# OpenReview forum: "Synthetic vs Real: Deep Learning on Controlled Noise"
_ICLR.cc/2020/Conference — Reject_

### Official Review · AnonReviewer1 · 2019-10-23
**Official Blind Review #1**

**Rating:** 3

**Review:**

This paper contributes a new dataset for testing label corruption robustness. The authors attempted to make the dataset represent "real-world noise." The hamartia is that they used the "search by image" feature from Google, so that "similar" images were what is determined "similar" by a convnet. In this way it is not clear their noise is any more real than what is seen in previous works (such as the Gold Loss Correction paper which used labels from a weak classifier as a source of label noise). The hamartia also makes their experimental findings and takeaways about "real noise" questionable, as it is not clear they are testing real noise but consequences of properties of convnet embeddings. However this paper still contributes a dataset, hence this paper still is some sort of contribution. However, with this flaw, it is not clear that it is enough for ICLR.

Small things:

> robust learning is experiencing a reminiscence in the deep learning era
renaissance?

>  ImageNet architectures generalize well on noisy data when the networks are fine-tuned. Comparing the first and second rows in Fig. 2, we observe that the test accuracy for fine-tuning is higher than that for training from scratch on both Red and Blue noise
This section should cite _Using Pre-Training Can Improve Model Robustness and Uncertainty_ (ICML 2019) since that is a main conclusion of their paper.

> Inception-ResNet-V2 (Szegedy et al., 2017) is used as the default network architectures
Unfortunately they're probably using some form of Inception to compute image similarities, which was how the "real noise" dataset was curated.

**Experience Assessment:**

I have published in this field for several years.

**Review Assessment: Checking Correctness Of Derivations And Theory:**

N/A

**Review Assessment: Checking Correctness Of Experiments:**

I assessed the sensibility of the experiments.

**Review Assessment: Thoroughness In Paper Reading:**

I read the paper at least twice and used my best judgement in assessing the paper.

---

> ### Author Response · Authors · 2019-11-15
> **Thank you for your comments and taking the time to review**
>
> We apologize that we have not made it clear how the noisy labels are constructed in our dataset. That seems to be the cause for R1’s misunderstanding. We appreciate that, despite the misunderstanding, R1 still acknowledges the contribution of our work.
>
>
> Q1: R1’s biggest concern is that our dataset includes noisy labels obtained from image-to-image search (“search by image”) which are determined by some unknown convnet embeddings. The image-to-image search noise may exploit the properties of the convnet, making experimental findings and takeaways “questionable”.
> Answer: Thanks for the comments. We have clarified this part in the revised paper.
>
> (1) The noisy images in our dataset are from the union of images independently retrieved from two sources (1) text-to-image and (2) image-to-image search, where the latter only accounts for a small proportion (28%) of the final dataset. We include image-to-image search results mainly to enrich the type of label noises in our dataset.
>
> (2) To resolve this concern, we exclude the images brought by image-to-image search from our dataset. This new data subset contains only the images from text-to-image search and thus is similar to noisy datasets in (Li et al., 2017a; Krause et al.,2016; Chen & Gupta, 2015). Please note these existing datasets do not have controlled noise.
>
> We re-conduct the experiments and include the results in A.3 in the Appendix (Figure 12, 13 and 14). We observe similar results that are consistent with what were reported in Section 5 (Figure 2, 3, and 4). To be specific, we confirm that our key findings still hold: (1) Deep Neural Networks (DNNs) generalize much better on real-world noise. (2) DNNs may not learn patterns first on real-world noisy data. (3) When networks are fine-tuned, ImageNet architectures generalize well on noisy data. In fact, the new ImageNet correlations are slightly higher. Rerunning the baseline comparison requires ~36,800 GPU hours (Tesla P100) or 4.2 GPU years. Given the limited time, we did not finish this part. We will provide the results in the future if needed.
>
> (3) For a qualitative comparison, we visualize the noisy examples in our dataset and compare them with unlabelled noisy training images in the WebVision benchmark. See Figure 1 and Figure 11 in the revised paper. We observe a similarity between the two datasets. Note WebVison does not provide the ground-truth labels and hence cannot be used in the controlled study. Adding manual labels to WebVision is not an option as its noisy images are highly imbalanced across classes.
>
> Inspired by R1’s comment, when releasing the dataset, we will make both types available: text-to-image search noisy data and image-to-image search noisy data.
>
>
> Q2: Experimental section should cite Using Pre-Training Can Improve Model Robustness and Uncertainty (ICML 2019).
> Answer: Thanks for the reference. We have cited this paper in the revised paper.
>
> Actually, this work, along with (Zhang et al, 2017), (Arpit et al. 2017) and (Kornblith et al., 2019) are good examples showing our results on synthetic noise are consistent with previous studies. Our contribution is the new results on the red noises which either reinforce or challenge the previous understandings about DNNs on noisy data.
>
>
> Q3: They're probably using some form of Inception to compute image similarities, which was how the "real noise" dataset was curated.
> Answer: We hope that we have clarified in the answer of Q1 that image-search noise would not be an issue in our study. If this question is about using inception in image deduplication, we run a similar CNN-based detector as in the previous work (Kornblith et al, 2019).
>
>
> Q4: Reminiscence → renaissance
> Thanks, we have fixed this typo.

---

### Official Review · AnonReviewer3 · 2019-10-24
**Official Blind Review #3**

**Rating:** 3

**Review:**

===========
Summary:
This paper introduces two real-world noisy datasets collected from google search based on two existing datasets: Mini-ImageNet and Stanford Cars. The author then conducted a series of experiments comparing 6 existing noisy label learning methods in two training settings: 1) from scratch, and 2) finetuning. Parameters were tuned when necessary. Based on the results, they made a few claims that challenges some of the previous believes in this field.

===========
My major concerns:
Collection new data using google search then run existing methods has limited contribution. And the new data are related to noisy LABELS only, which does not cover any input noise such as low resolution, abnormal size, black/blank/carton background etc.
The web data searched are of moderate or rather small scales, which doesn’t seem to contribute much to the community. As mentioned in the paper, we already got some real-world datasets such Clothing-1M.
How the new datasets are collected, labeled and mixed are not clear. The numbers do not seem to add up. In text, the authors said they collected 94906/51687 images with annotations. However, in Table 1, there are 39000/8144 in datasets Red Mini-ImageNet/Red Standford Cars. The last paragraph on Page 3, how exactly the negative examples come from? How the “negative” is defined, since you got more than 1 nanotations for each image. “ Following the construction of synthetic datasets, we replace the training images in the original dataset”, refers to which original dataset? The Mini-ImageNet/Red Standford Cars or the web-searched? I don’t quite get it which dataset is creating here. Are the Red noise datasets a mixture of Mini-ImageNet/Red Standford Cars and web-search images, or are they only web-searched images.
The findings are questionable, due to the specific way how the web data are collected. Since the web datasets are created by similarity search based on a pool of seed images (5000), all the searched images are subject to the small number of seed images and the similarity algorithm of google search. Although some filtering was done to reduce test vs training duplication, the entire dataset can be treated as a duplicate of the 5000 seed images, which is roughly 10% of Mini-ImageNet -- the intrinsic noise rate is low. Having this in mind, some of the findings become no surprise at all, for example, 1) DNNs are robust to real-world noise, 2) performance does not decrease much as training progresses, 3) real-world noise is less harmful, and 4) DNN performance does not change much with different robust methods. These are all phenomena of low noise rates.

**Experience Assessment:**

I have published in this field for several years.

**Review Assessment: Checking Correctness Of Derivations And Theory:**

I assessed the sensibility of the derivations and theory.

**Review Assessment: Checking Correctness Of Experiments:**

I carefully checked the experiments.

**Review Assessment: Thoroughness In Paper Reading:**

I read the paper thoroughly.

---

> ### Author Response · Authors · 2019-11-15
> **Thanks for the review. We have clarified an important misunderstanding (Part 2/2)**
>
> Q2: How the new datasets are collected, labeled and mixed are not clear.
>
> Q2.1 The numbers do not seem to add up 94906/51687 in the text but 39000/8144 in Table 1?
> Answer: The numbers are correct. In Table 1, our Red datasets only use the images of false labels out of the total 94906/51687 annotated images. Hence the numbers are smaller. We also create two datasets by adding all annotated images to the original datasets (called “Augmented” in Table 1), the size of which are 144906=94906+50000 and 59627=51687+8144-204(duplicates). Note we detect 381 near-duplicated images in Mini-ImageNet and 204 in Stanford Cars.
>
> Q2.2 How exactly the negative examples come from? How the “negative” is defined, since you got more than 1 nanotations for each image?
> Answer: the negative examples are the union of images independently retrieved from two sources (1) text-to-image and (2) image-to-image search. We manually annotate every retrieved image to identify the ones with false labels as negative examples. It was stated in our paper “the final label is reached by majority voting”.
>
>
> Q2.3: Following the construction of synthetic datasets, we replace the training images in the original dataset”, refers to which original dataset? The Mini-ImageNet/Red Stanford Cars or the web-searched? Are the Red noise datasets a mixture of Mini-ImageNet/Red Standford Cars and web-search images, or are they only web-searched images?
> Answer: It refers to the standard Mini-ImageNet and Stanford Cars benchmark.
>
> Red Mini-ImageNet is a mixture of Mini-ImageNet and web-searched images. See Figure 10 in the Appendix. As stated in the revised Section 3, our dataset is constructed in the same way as the synthetic dataset with only one difference, i.e. we draw noisy examples from web search images. Since Mini-ImageNet and Stanford Cars are all collected from the web, their true positive examples should follow a similar distribution as the added noisy images.  We have experimental results on web-search only images, where we have a similar observation. Due to the space limit, we did not include them.
>
>
> Q3: The web data searched are of moderate or rather small scales, which doesn’t seem to contribute much to the community….we already got some real-world datasets such Clothing-1M?
> Answer: We hope the difference is now clear. Existing real-world datasets such as Clothing-1M or WebVision do not provide ground-truth labels for every training example. Their noise level is thus fixed and unknown, making them unsuitable for controlled studies. That explains why existing controlled studies were all done on synthetic datasets.
>
> We create the first controlled dataset of web search noisy images to enable future research. We make our data size to be comparable to existing ones to study their difference.
>
> Note that even though WebVision may contain labeled positive images (used for evaluation), these positive images are not useful in creating controlled noise where labeled images with false labels are needed. Adding manual labels to WebVision is not an option as its noisy images are highly imbalanced across classes.
>
>
> Q4: The new data are related to noisy label only, which does not cover any input noise such as low resolution, abnormal size, black/blank/carton background etc.
> Answer: Our dataset includes Black/Blank/Carton background. See Figure 1 in the revised paper.
> Low resolution, abnormal size, and other image corruption have been studied in ( https://openreview.net/forum?id=HJz6tiCqYm ) and are beyond the topic of our study.
>
> We refer image-search noise as “real-world noise” to distinguish it from synthetic label-flipping noise and have made
> it clear in the revised paper.
>
>
> Q5: Collection new data using google search then run existing methods has limited contribution.
> Answer: We respectfully disagree with this statement. Our contribution is twofold. First, we establish a dataset of controlled real image search noise. This is acknowledged by R1 and R2.
>
> Beyond the dataset, our contribution improves our community's knowledge of deep learning on noisy data. Specifically, we conducted the largest study (to date) into understanding DNN training on noisy data across a variety of noise levels and types, architectures, methods, and training settings. Our study confirms existing findings in (Zhang et al, 2017), (Arpit et al. 2017) and (Kornblith et al., 2019) on synthetic noise, and furthermore brings forward new findings that either reinforce or challenge the previous understandings about DNNs on noisy data. These findings are the result of conducting ~3,000K experiments using ~60,000 GPU hours (Tesla P100) or 7 GPU years. It is worth noting that in the studies (Zhang et al, 2017), (Arpit et al. 2017) and (Kornblith et al., 2019)  and many other ones, they all ran “existing” methods.
>
> With the clarified data construction procedure, we hope R3 can take this into account when making the final decision.

---

> ### Author Response · Authors · 2019-11-15
> **Thanks for the review. We have clarified an important misunderstanding (Part 1/2)**
>
> From R3’s comments, we believe there is a serious misunderstanding about our dataset. It seems that R3 believes our dataset is collected from a small number of (5000) seed images using Google’s image similarity search. As a result, R3 thinks the intrinsic noise rate is low which may question our findings.
>
> We understand R3’s concern. In fact, we would have shared similar ones if the dataset had been constructed in the way R3 has interpreted the paper. However, we must point out that this is a misunderstanding.
>
> Below, we will address all questions from R3. To start, we review how our dataset is constructed. The paragraphs below are quoted from the revised Section 3 with references removed for clarity.
>
> “To recap, let us revisit the construction of existing noisy datasets in the literature. For the real-world noisy datasets, one automatically collects images for a class by matching the class name to the images' surrounding text. The retrieved images include false positive (or noisy) examples, i.e. text match/search thinks an image is a positive when it is not. As their training images are not manually labeled, the data noise level is fixed and unknown. As a result, these datasets are unsuitable for controlled studies.
>
> On the other hand, the synthetic noisy dataset is built on the well-labeled dataset. The label of each training example is independently changed to a random incorrect class with a probability p, called noise level, which indicates the percentage of training examples with false labels. Since the ground-truth labels for every image are known, previous studies enumerate p to obtain datasets of different noise levels and use them in controlled experiments. On balanced datasets, such as Mini-ImageNet and Stanford Cars used in our study, the above process is equivalent to first sampling p% training images from a class and then replacing them with the images uniformly drawn from other classes. The drawback is that their noisy labels are artificial and do not follow the distribution of the real-world noise.
>
> For our dataset, we follow the construction of synthetic datasets with only one difference, i.e. we draw noisy (false positive) examples from similar noise distributions as in existing real-world noisy datasets. To be specific, we draw images using Google text-to-image search, which is commonly used to get noisy labels in prior works. In addition, we also include noisy examples from  image-to-image search to enrich the type of label noises in our dataset. We manually annotate every retrieved image to identify the ones with false labels. For each class, we replace p% training images in the Mini-ImageNet/Stanford Cars dataset with these false-positive images. We enumerate p in 10 different levels... The constructed datasets hence contain label noise similar to existing real-world datasets and are suitable to be used in controlled experiments.”
>
>
> Q1: The findings are questionable, due to the specific way how the web data are collected.  Since the web datasets are created by similarity search based on a pool of seed images (5000) … the entire dataset can be treated as a duplicate of the 5000 seed images, which is roughly 10% of Mini-ImageNet -- the intrinsic noise rate is low.
> Answer: We respectfully point out that this is a misunderstanding. Our annotated images come from the union of images independently retrieved from two sources (1) text-to-image and (2) image-to-image search, where the latter only accounts for ~28% of the total dataset.
>
> There are no seeds images in any of the search processes.
> In the text-to-image search, we select the top 5,000 images but do not use them as the seed.
> In the image-to-image search, we do not use seed images either. We use every single training image in Mini-ImageNet (i.e. 50,000) and Stanford cars (i.e. 8,144) as the query. That is we issue 50,000 image queries using 100% of Mini-ImageNet training images. As a result, the entire dataset is not a duplicate from 10% seed images and the intrinsic noise rate is not low.
>
> We visualize the noisy examples in our dataset and compare them with the noisy training images in the WebVision benchmark. See Figure 1 and Figure 11 in the revised paper. We observe a similarity between the two datasets. Note that WebVison does not provide the ground-truth labels and hence cannot be used in the controlled study.
>
> To resolve the concern about image-to-image search, we exclude the images brought by image-to-image search from our dataset. This new data subset contains only the images retrieved from the text-to-image search and thus is similar to noisy datasets in previous works (Li et al., 2017a; Krause et al.,2016; Chen & Gupta, 2015). We re-conduct most of the experiments and include the results in A.3 in the Appendix (Figure 12, 13 and 14). We observe similar results that are consistent with what were reported in Section 5 (Figure 2, 3, and 4).

---

### Official Review · AnonReviewer2 · 2019-10-24
**Official Blind Review #2**

**Rating:** 6

**Review:**

Performing controlled experiments on noisy data is essiential in understanding deep learning. But there is lack of suitable data. In this paper, the authorsestablish a large benchmark of controlled real-world noise, which is one contribution of it. For this paper, the dataset they established is a main contribution. There is no much contributions in methods.

Based on the dataset and previous works, the study find something. for example, DNNS generalize much better on real-world noise, and DNNs may not learn paterns first on real-world noisy data, and so on.

**Experience Assessment:**

I have read many papers in this area.

**Review Assessment: Checking Correctness Of Derivations And Theory:**

N/A

**Review Assessment: Checking Correctness Of Experiments:**

I carefully checked the experiments.

**Review Assessment: Thoroughness In Paper Reading:**

I read the paper thoroughly.

---

> ### Author Response · Authors · 2019-11-15
> **Thanks for the review**
>
> We thank R2 for the positive reviews and for acknowledging the contribution of our dataset and findings.
>
> We agree that our main contribution is not in methods.
>
> Regarding the method, we have shown how large a difference hyperparameter tuning can make in existing methods as each needs to be run on a spectrum of noise levels. We show that by extensively searching hyperparameters,  our implementation of  (Jiang et al. 2018) and (Zhang et al. 2018) gets up to a 2.9% gain over what was reported in the papers, achieving a performance that is comparable to, or even better than, the state-of-the-art methods (Shu et al., NeurIPS2019) and (Arazo et al. ICML 2019). See A.2 in the Appendix. With this knowledge, we have run a large number of experiments (1,840) to faithfully compare the existing methods. This lays the foundation for our findings about these robust learning methods and creates a baseline for future research to compare.
>
> Beyond the dataset, our contribution improves our community's knowledge of deep learning on noisy data. Specifically, we conducted the largest study (to date) into understanding DNN training on noisy data across a variety of noise levels and types, architectures, methods, and training settings. Our study confirms existing findings in (Zhang et al, 2017), (Arpit et al. 2017) and (Kornblith et al., 2019) on synthetic noise, and furthermore brings forward new findings that either reinforce or challenge the previous understandings about DNNs on noisy data. These findings are the result of conducting ~3,000K experiments using ~60,000 GPU hours (Tesla P100) or 7 GPU years.

---

### Decision · Program_Chairs · 2019-12-19

**Decision:**

Reject

**Comment:**

Thanks for your detailed feedback to the reviewers, which helped us a lot to better understand your paper.
However, given high competition at ICLR2020, we think the current manuscript is premature and still below the bar to be accepted to ICLR2020.
We hope that the reviewers' comments are useful to improve your manuscript for potential future submission.